# Semaphorin 3A: A potential target for prevention and treatment of nickel allergy

Lipei Liu [1], Megumi Watanabe [1✉], Norikazu Minami[1], Mohammad Fadyl Yunizar [1] & Tetsuo Ichikawa [1]

Metal allergy is one of the typical immune disorders encountered during the application of dental/medical materials and has a highly complex pathogenic mechanism. Semaphorin 3A (Sema3A), a member of the semaphorin family, is reported to be involved in various immune disorders. However, its role in metal allergy has not been clarified yet. Herein, we show that Sema3A expression was upregulated in nickel (Ni) allergy-induced mouse ear tissue and in $NiCl_2$-stimulated mouse keratinocytes. Moreover, Sema3A regulated tumor necrosis factor-alpha production and mitogen-activated protein kinase activation in keratinocytes. The specific deletion of Sema3A in keratinocytes did not affect immune cell infiltration but reduced edema and ear swelling; it also impeded Th1 responses to cause a slight alleviation in Ni allergy in mice. Our results demonstrate that Sema3A promotes the development of metal allergy and should be explored as a potential target for the prevention and treatment of metal allergy.

[1] Department of Prosthodontics & Oral Rehabilitation, Tokushima University, Graduate School of Biomedical Sciences, 3-18-15, Kuramoto, Tokushima 770-8504, Japan. ✉email: megwat@tokushima-u.ac.jp

Metal allergy, one of the typical disorders of the immune system, is a major challenge in the application of dental/ medical materials[1]. It is estimated that up to 17% of women and 3% of men are allergic to nickel (Ni) and that 1–2% of individuals are allergic to cobalt, chromium, or both[2]. Metal allergy not only causes treatment failure but also affects the skin throughout the body; it has been reported to be associated with lichen planus lesions and palmoplantar pustulosis[3–5]. An acidic diet, decomposition of food residues, and bacterial metabolism in the oral cavity result in chemical corrosion of metal materials[6]. Corroded and ionized metals tend to cause metal allergy; however, the mechanism is highly complex, and many puzzles remain to be solved.

It takes repeated or prolonged exposure to develop a metal allergy. Metal ions released from various alloys that come in contact with the skin could activate epithelial cells, such as keratinocytes, to produce cytokines or chemokines as well as activate antigen-presenting cells (APCs), such as dendritic cells (DCs) and Langerhans cells. After APCs present antigens to naïve T cells in the draining lymph nodes, the differentiated effector T cells migrate to the peripheral tissues and lead to allergic reactions[7].

Ni, among various metals, is the most common allergen that causes metal allergy[8]. We successfully built a Ni allergy mouse model in our previous study[9–11]. Using this model, we demonstrated that the activation of DCs through p38 MAPK/MKK6 is important for the development of Ni allergy in mice[9] and that the thymic stromal lymphopoietin (TSLP)/TSLP receptor-mediated interaction between epithelial and immune cells could trigger Ni allergy[10]. We have also demonstrated that semaphorin 7A, a member of the semaphorin family, enhances inflammation during the effector phase of Ni allergy[11].

Semaphorins are a type of secreted and membrane proteins that were initially recognized as axon guidance molecules. Accumulating evidence suggests that semaphorins are involved in the pathogenesis of several immune disorders[12]. Semaphorin 3A (Sema3A), a secretory member of the semaphorin family, is known to function as an effective immunomodulator in immune cell migration and regulation[13–17]. Sema3A downregulates T cell proliferation via the neuropilin-1 (Nrp1)-plexin-A4 receptor complex[17] and increases cytokine production in response to toll-like receptor (TLR) agonists or bacterial sepsis in mice[18]. Sema3A is also involved in DC transmigration from the periphery to draining lymph nodes[17]. Besides, Sema3A has been associated with various immune diseases, such as rheumatoid arthritis and systemic lupus erythematosus[19]. It also controls the inflammatory responses in allergic conjunctivitis and allergic rhinitis[20,21]. Recent studies have suggested that Sema3A could interrupt the itch-scratch cycle by inhibiting the extension and sprouting of C-fiber in the epidermis, thereby alleviating the skin lesions and scratching behavior of NC/Nga mice, a model of atopic dermatitis (AD), in a dose-dependent manner[22,23]. Considering its role as a potent inhibitor of neurite outgrowth of sensory neurons and inflammatory cell infiltration, Sema3A has been extensively studied for its potential in the treatment of refractory itching skin diseases. The exogenous application of Sema3A has been shown to alleviate the pathophysiology of AD and psoriasis[24]. As metal allergy results in contact dermatitis, a chronic skin condition characterized by inflammation similar to AD, we hypothesized that there is an association between Sema3A and metal allergy, which has not been reported.

To better clarify the molecular mechanism underlying metal allergy, in this study, an Ni allergy mouse model was used to study the role of Sema3A in Ni allergy. We speculated that Sema3A, being a secretory protein, might function like cytokines or chemokines and may play a crucial role in both intercellular and intracellular communication, thereby influencing the process of metal allergy

development. Furthermore, it might be a potential target for the investigation and treatment of metal allergy.

## Results

**Sema3A expression is upregulated in keratinocytes upon stimulation with NiCl$_2$.** The epidermis, which is the outermost layer of the skin, is the first barrier against antigen invasion and plays vital roles in the development of metal allergy[25,26]. We aimed to determine whether there is any interaction between Sema3A and keratinocytes, which constitute 90% of the cells of the epidermis, thereby contributing to metal allergy. To investigate Sema3A expression in keratinocytes, the Pam2.12 keratinocyte cell line was stimulated with NiCl$_2$ for different durations. The Sema3A mRNA expression was upregulated in keratinocytes exposed to NiCl$_2$ stimulation over time, although the difference was not significant when compared with the expression in the control (0 h time point) (Fig. 1a). The expression of Sema3A protein was also upregulated and reached a peak at 12 h after NiCl$_2$ stimulation, as indicated by western blotting (Fig. 1b). At 72 h after NiCl$_2$ stimulation, the Sema3A protein levels were significantly increased in the culture medium of Pam2.12, as confirmed by ELISA (Fig. 1c). The staining of keratinocytes with Sema3A antibody showed brightly positive cytoplasmic secretory granules that filled the entire cell cytoplasm. These positive granules were observed to be more strongly expressed on keratinocytes at 24, 48, and 72 h after NiCl$_2$ stimulation compared with those in the control group. The quantification analysis using BZ-X800 Analyzer software showed that the Sema3A-positive area was significantly greater in the 24 and 48 h groups than in the control group (Fig. 1d and Supplementary Fig. 1). Additionally, the viability of Pam2.12 cells was tested after stimulation with NiCl$_2$ for different durations using Readidrop$^{TM}$ Propidium Iodide in flow cytometry. Results showed that the cell viability was not affected by this concentration of Ni, thereby eliminating the possibility that the increased Sema3A production was due to cell death (Fig. 1e).

**Sema3A knockdown in keratinocytes alters MAPK activation and TNF-α expression.** To evaluate the functional role of Sema3A in keratinocytes, Pam2.12 cells were transfected with Sema3A siRNA. Without NiCl$_2$ stimulation, the expression of TNF-α, a known inflammatory cytokine produced by keratinocytes[7], showed no significant difference among groups. However, when cells were stimulated with NiCl$_2$, the NiCl$_2$-stimulated control and mock group showed higher TNF-α expression than the non-stimulated groups. The expression of TNF-α in the Sema3A-silenced group was significantly inhibited compared with that in the mock group 24 h after NiCl$_2$ stimulation. The cells seemed to recover at 48 h post-siRNA transfection and exhibited less efficient suppression of TNF-α mRNA than that at 24 h post-siRNA transfection (Fig. 2a). TNF-α production in the Sema3A-silenced group was also significantly inhibited compared with that in the control group (Fig. 2b). MAPK activation was also assessed in Pam2.12 cells. In Sema3A-silenced cells, the activation of p38 was significantly inhibited at 24 h after NiCl$_2$ stimulation compared with that in the control. The cells seemed to recover from the suppression effect of the siRNA at 48 h after stimulation, although the p38 protein expression was still lower than that in the control group (Fig. 2c). These results suggest that Sema3A may play a role in triggering or regulating the process of Ni allergy through p38 kinase activation and TNF-α expression in keratinocytes.

**Sema3A is enhanced in Ni allergy-induced mouse ear tissue.** To investigate Sema3A expression in vivo, Ni allergy in mice was induced as described in Fig. 3a. After 48 h of Ni re-challenge,

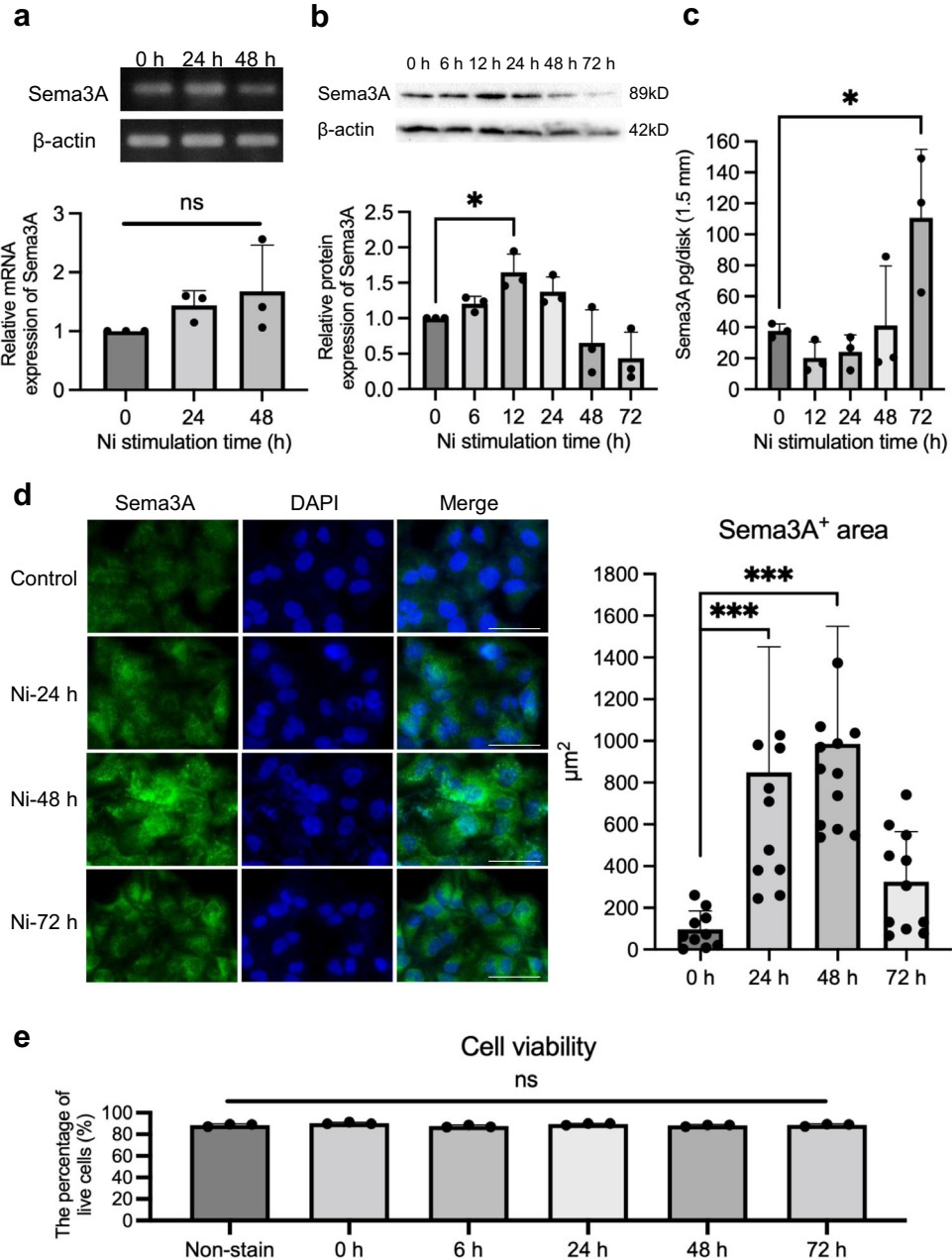

**Fig. 1 Ni stimulation upregulates Sema3A expression in Pam2.12 cells. a** Sema3A was expressed on $NiCl_2$-stimulated Pam2.12 cells and its mRNA levels increased after $NiCl_2$ stimulation. Gene expression was normalized to levels of β-actin. $N = 3$. **b** Upregulated Sema3A in $NiCl_2$-stimulated Pam2.12 cells analyzed by western blotting. β-actin was used as the internal control. $N = 3$. The uncropped Western blotting image is shown in Supplementary Fig. 5a. **c** Upregulated Sema3A production in $NiCl_2$-stimulated Pam2.12 cells analyzed by ELISA. $N = 3$. **d** Immunocytochemistry of Sema3A expressed in Pam2.12 cells 24, 48, and 72 h after $NiCl_2$ stimulation. Scale bar, 50 μm. Quantification of the area of Sema3A-positive Pam2.12 cells (at least 10 images/group) after Ni stimulation analyzed by BZ-X800 Analyzer software. **e** The population of live Pam2.12 cells after stimulation with 250 μM Ni for different durations analyzed by flow cytometry using propidium iodide staining. $N = 3$. Data are shown as mean ± SD and are representative of at least three independent experiments. Different letters on bars indicate significant differences. *$p < 0.05$, ***$p < 0.001$; One-Way ANOVA with LSD. Supplementary Data and $p$ values are provided as a Supplementary Data file.

delayed-type hypersensitivity (DTH) reaction, characterized by severe swelling and redness in the ear tissue, was confirmed in the allergy-induced group (Fig. 3b). Hematoxylin and eosin (H&E) staining showed increased thickness and more infiltrated cells in the allergy-induced ear tissue (Fig. 3c). Ear thickness was significantly increased 24, 48, and 72 h after $NiCl_2$ injection compared with that in the control group (Fig. 3d). Sema3A expression was found to be mainly located in the epidermis layer and expressed more strongly than that in the control. The

quantification analysis using BZ-X800 Analyzer software showed a significantly larger Sema3A-positive area in the Ni-stimulated group than in the control group (Fig. 3e and Supplementary Fig. 2). Consistently, the western blotting results of Sema3A showed significantly stronger expression in the Ni allergy group than in the control (Fig. 3f). Inflammatory cytokines and chemokines associated with skin inflammation were also investigated. The upregulation of TNF-α, IL-1β, L-23, CXCL1, and CCL20 mRNA was observed in Ni allergy-induced mouse ear

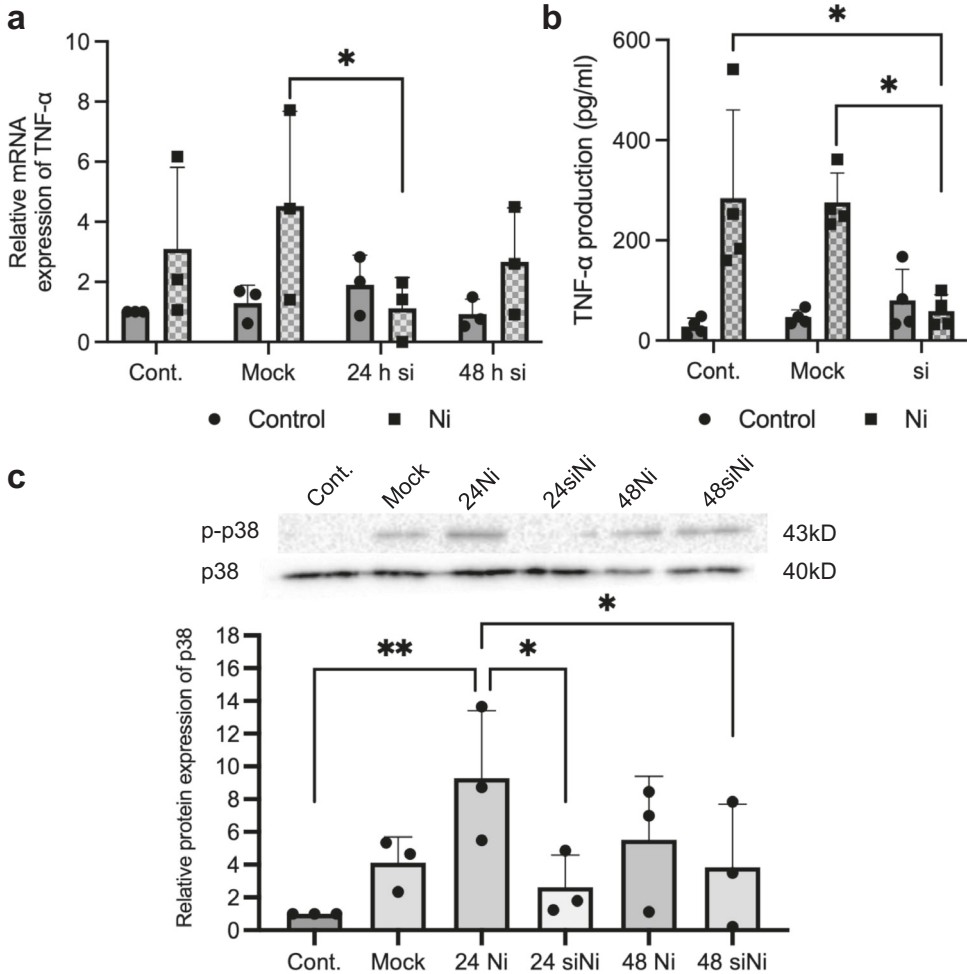

**Fig. 2 Sema3A-silenced Pam2.12 cells show suppressed TNF-α expression and MAPK activation upon Ni stimulation. a** TNF-α mRNA expression in Sema3A siRNA-treated Pam2.12 cells after NiCl₂ stimulation analyzed using quantitative RT-PCR. Gene expression was normalized to levels of β-actin. 24 h si/48 h si represents Sema3A siRNA-treated groups with or without 24/48 h of NiCl₂ stimulation. $N = 3$. **b** TNF-α protein production in Sema3A siRNA-treated Pam2.12 cells after NiCl₂ stimulation analyzed by ELISA. The samples with or without NiCl₂ stimulation were collected 24 h after stimulation. $N = 4$. **c** Inhibited p38 phosphorylation in Sema3A siRNA-treated Pam2.12 cells after NiCl₂ stimulation analyzed by western blotting analysis. 24 Ni/48 Ni represents the groups after 24/48 h of NiCl₂ stimulation. 24 siNi/48 siNi represents the Sema3A siRNA-treated groups after 24/48 h of NiCl₂ stimulation. $N = 3$. The uncropped Western blotting image is shown in Supplementary Fig. 5b. Data are shown as mean ± SD and are representative of at least three independent experiments. *$p < 0.05$, **$p < 0.01$; One-Way ANOVA with LSD. Supplementary Data and $p$ values are provided as a Supplementary Data file.

tissues (Fig. 3g). As Sema3A expression was strongly enhanced in the Ni allergy-induced mouse ear tissue than in the control, it is speculated that Sema3A is positively correlated with the development of Ni allergy.

**The number of T cells, macrophages, and DCs increases in Ni allergy-induced mouse ear tissue**. Results of flow cytometry showed that the number of CD4⁺ and CD8⁺ T cells increased markedly in the Ni allergy-induced groups (Fig. 4a, b). Notably, compared with the number of CD3⁺CD8⁺ T cells, the number of CD3⁺CD4⁺ T cells significantly increased in the Ni allergy-induced mouse ears (Fig. 4b). The gating strategy of macrophages and DCs in Ni allergy-induced mouse ears is shown in Fig. 4c. Both macrophages and DCs increased in number in the Ni allergy-induced mouse ear tissue (Fig. 4d). In the immuno-fluorescence staining of frozen sections, an increased number of CD4⁺ and CD8⁺ T cells was observed in the Ni allergy-induced ear tissue (Fig. 4e, f). Similarly, an increase in the number of CD11b⁺F4/80⁺ macrophages was observed in the Ni allergy-induced group compared with that in the control group (Fig. 4g). The immunofluorescence staining of epidermal sheet showed that

the number of CD11c⁺MHC class II⁺ DCs increased in the Ni allergy-induced group compared with that in the control group (Fig. 4h). The quantification results of these immunofluorescence images using BZ-X800 Analyzer showed a significant increase in the number of CD4⁺ T cells, CD11b⁺F4/80⁺ macrophages, and CD11c⁺MHC class II⁺ DCs in the Ni allergy-induced group compared with that in the control group; however, the number of CD8⁺ T cells showed no difference in the two groups (Fig. 4i and Supplementary Fig. 3). These results indicate that CD4⁺ T cells, CD11b⁺ macrophages, and CD11c⁺ DCs, which are responsible for immune response regulation and antigen representation, may play important roles in the development of Ni allergy.

**Sema3A-specific deletion in keratinocytes reduces Ni allergy**. Sema3A is reported to mainly localize in keratin14-positive keratinocytes in the stratum basale of the human epidermis[27]. Therefore, to investigate the role of Sema3A in Ni allergy in vivo, Sema3A^fl/fl mouse was crossed with keratin5 (K5)-Cre mouse using a cre/loxp system to knockout Sema3A in K5, which is a similar protein and partner of keratin14. The DTH reaction was confirmed in Sema3A^fl/fl mice, similar to that in C57BL/6J mice

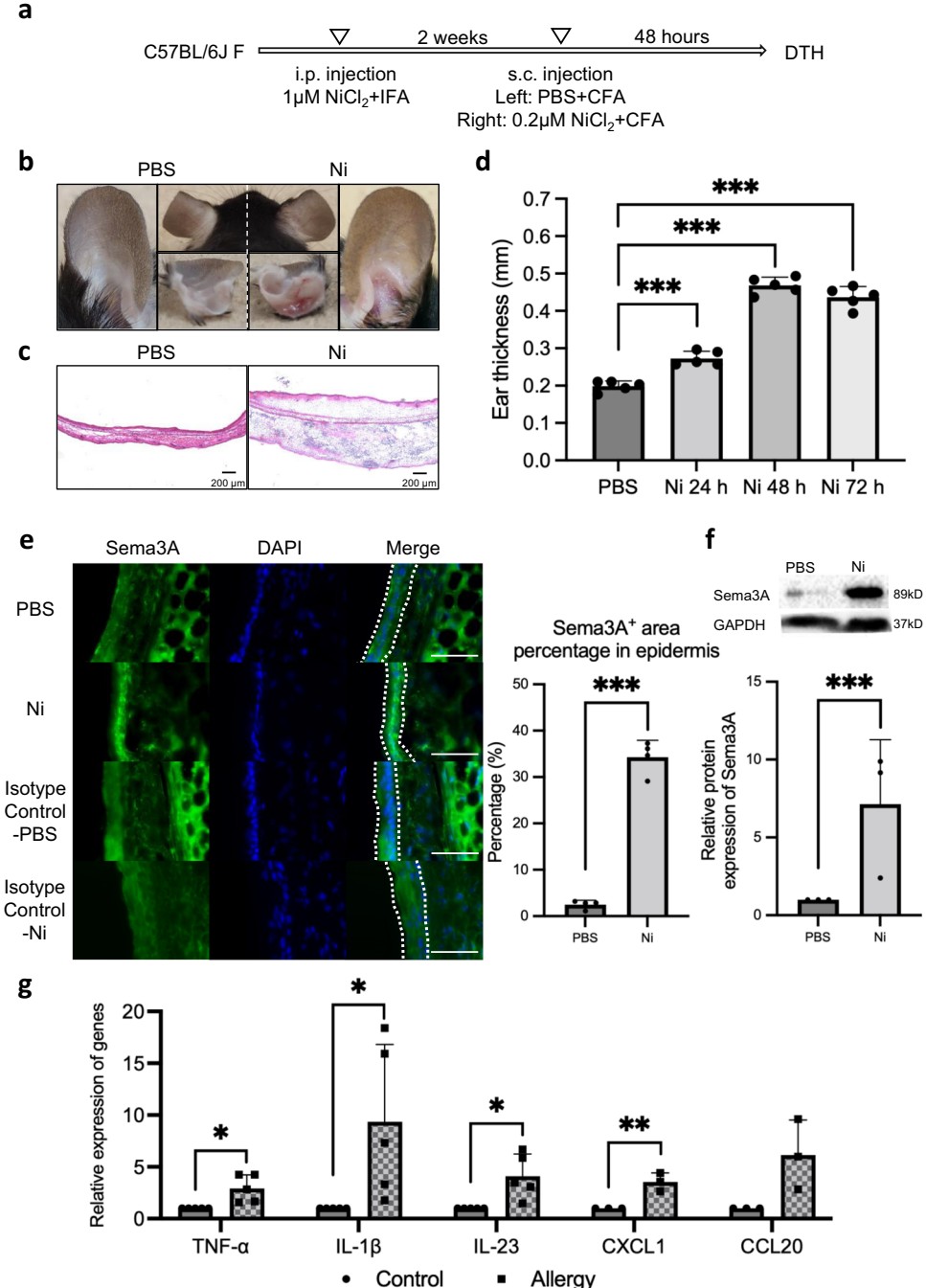

**Fig. 3 Sema3A expression is upregulated in the Ni allergy site. a** Protocol for inducing Ni allergy in mouse ear tissue. **b** Representative photos of the Ni allergy-induced mouse ear and the control ear. Photos show the ventral side (right and left), dorsal side (middle-upper), and cross-section (middle lower). **c** Histological images of Ni allergy-induced mouse ear tissue and control ear tissue. Scale bar, 200 μm. **d** DTH was determined by measuring ear thickness after Ni re-challenge. Data are shown as mean ± SD of 5 mice/group. **e** Sema3A expression in the control and Ni allergy-induced ear tissue stained via an immunofluorescence assay. Staining with only anti-rabbit Fluor Alexa 488 secondary antibody was set as isotype controls for both PBS control group and Ni-challenged group. Dashed line shows the epidermis zone. Scale bar, 50 μm. The percentage of the Sema3A-positive area (4 images/group) in the total area of epidermis in the ear tissue was analyzed using the BZ-X800 Analyzer software. **f** Upregulated Sema3A expression in Ni allergy-induced mouse ear tissue analyzed by western blotting analysis. GAPDH was used as the internal control. $N = 3$. The uncropped Western blotting image is shown in Supplementary Fig. 5c. **g** The mRNA expression of inflammatory cytokine/chemokine in Ni allergy-induced C57BL/6J mouse ear tissue analyzed using quantitative RT-PCR. Gene expression was normalized to levels of β-actin. $N = 5$. Data are shown as mean ± SD and are representative of at least three independent experiments. $*p < 0.05$, $**p < 0.01$, $***p < 0.001$; Two independent samples $t$-test. Supplementary Data and $p$ values are provided as a Supplementary Data file.

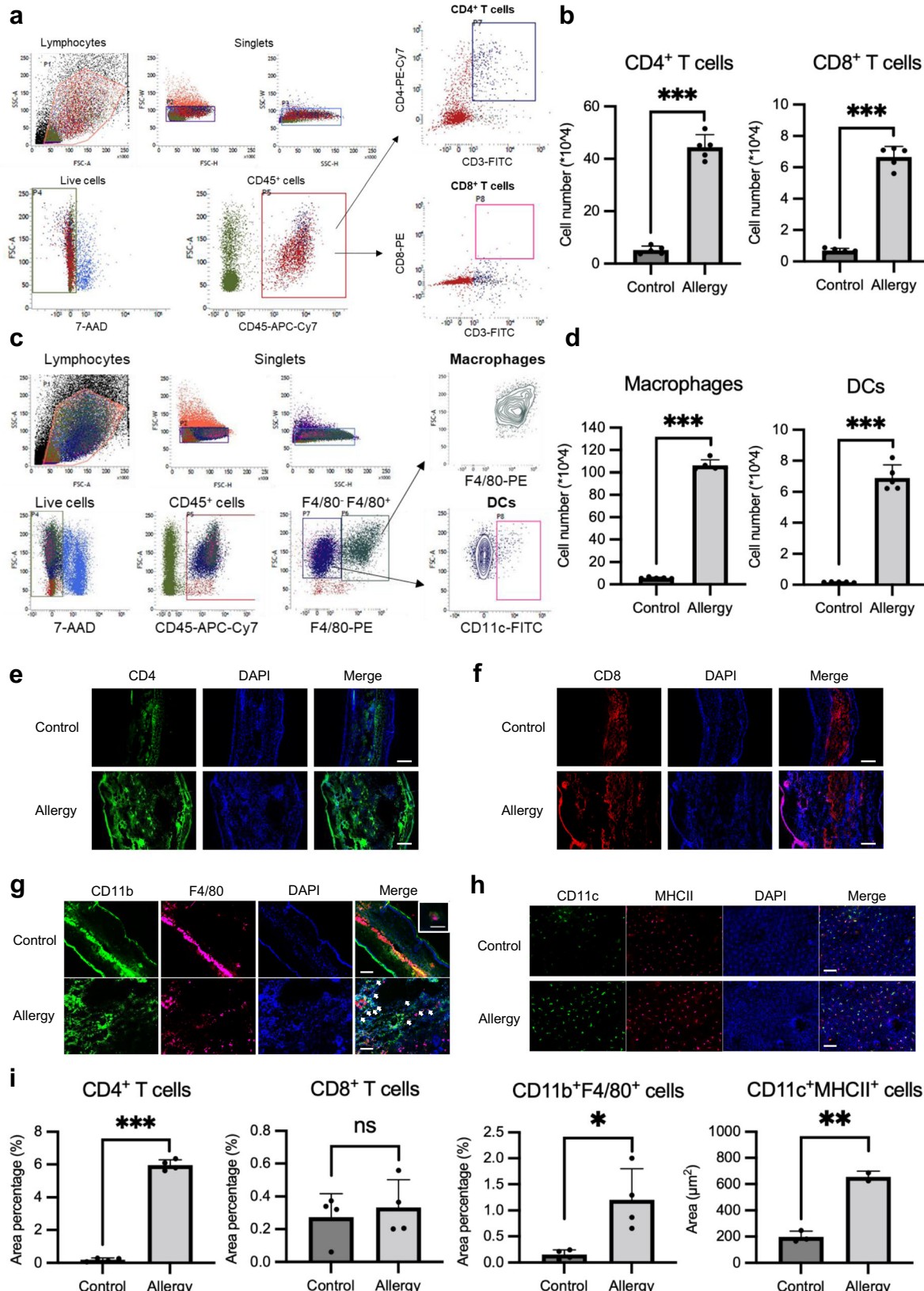

(Supplementary Fig. 4); therefore, Sema3A$^{fl/fl}$ mice was used as a control group in the confirmation of Ni allergy development in Sema3A conditional knockout (cKO) mice. Results showed that the development of Ni allergy was alleviated in Sema3A cKO mice compared with that in the control mice (Fig. 5a, b). After 24 h of Ni elicitation, the ear thickness of the control mice was

significantly increased in the Ni allergy-induced ear tissue compared to that in the control tissue, while the ear thickness of Sema3A cKO mice only slightly increased following Ni allergy induction and not to the degree observed in the control group (Fig. 5a). After 48 h, the ears of the control mouse remained red and swollen; in contrast, although the ears of the Sema3A cKO

**Fig. 4 The number of immune cells increases at the site of Ni allergy. a** Representative flow cytometry analysis of CD4[+] T cells and CD8[+] T cells in mouse ears 48 h after Ni challenge. **b** The number of CD4[+] T cells and CD8[+] T cells in the Ni allergy-induced mouse ear tissue. Data are shown as mean ± SD of 5 mice/group and are representative of three independent experiments **c** Representative flow cytometry analysis of macrophages and DCs in mouse ears at 48 h after Ni challenge. **d** The number of macrophages and DCs in the Ni allergy-induced mouse ear tissue. Data are shown as mean ± SD of 5 mice/group and are representative of three independent experiments. CD4[+] T cells (**e**) and CD8[+] T cells (**f**) in the control and Ni allergy-induced ear tissue were stained using immunofluorescence assays. Scale bar, 100 μm. **g** CD11b[+]F4/80[+] macrophages in the control and Ni allergy-induced ear tissue were stained using immunofluorescence assays. Arrows indicate double-positive cells. Scale bar, 100 μm. Higher-magnification images of CD11b[+]F4/80[+] macrophages are shown in the upper right of the control group images. Scale bar, 10 μm. **h** CD11c[+]MHC class II[+] DCs in the epidermal sheet of mouse ears analyzed using an immunofluorescence assay. Scale bar, 100 μm. **i** The area percentage of CD4[+] T cells (4 images/group), CD8[+] T cells (4 images/ group), CD11b[+]F4/80[+] macrophages (4 images/group) in the total area of ear tissue, and the area of the CD11c[+]MHC class II[+] DCs (3 images/group) analyzed using the BZ-X800 Analyzer software. *$p < 0.05$; **$p < 0.01$, ***$p < 0.001$; Two independent samples $t$-test. Supplementary Data and $p$ values are provided as a Supplementary Data file.

mice remained red, the ear swelling had subsided (Fig. 5b). The ratio of Ni allergy-induced ear thickness to the control ear thickness, which represents the degree of thickening of ears, was significantly lower in the Sema3A cKO group than in the control group 24 h and 48 h post-Ni re-challenge (Fig. 5a). In other words, when inducing Ni allergy, the ear swelling in Sema3A cKO mice was less severe than that in the control mice. In the H&E staining of Ni allergy-induced ear tissue of Sema3A cKO mice, infiltrated inflammatory cells could be observed, but the edema was less severe than in the control mice (Fig. 5c). To determine the genes, cytokines, and chemokines associated with Sema3A deletion in keratinocytes, the expression levels of TNF-α, IL-1β, IL-23, and CXCL1 were investigated. The expression of these genes was found to be significantly enhanced in the Ni allergy-induced control mouse ear tissue but suppressed in the Sema3A cKO group. The upregulation of CCL20 was likely unaffected with the deletion of Sema3A (Fig. 5d). Further verification of the mRNA expression of related anti-inflammatory cytokines in Ni allergy-induced mouse ear tissue showed that the expression levels of IL-6, IL-10, and IL-13 were significantly higher in the Ni allergy-induced Sema3A cKO mouse ear tissue than in the control group (Fig. 5e). These results indicate that Sema3A in keratinocytes probably promotes the development of Ni allergy by altering ear swelling and the production of inflammatory and anti-inflammatory cytokines.

**Sema3A-specific deletion in keratinocytes increases cell infiltration in the allergy site.** Flow cytometry was performed to examine the change in immune cell profiles during Ni allergy development in Sema3A cKO mice. The number of CD4[+] and CD8[+] T cells was found to be increased in the Ni allergy-induced Sema3A cKO mouse ears compared to that in the Ni allergy-induced Sema3A[fl/fl] mouse ears. However, the CD4[+]/CD8[+] T cell ratio in the allergy groups showed no difference between the two types of mice (Fig. 6a). The number of macrophages was increased in the Ni allergy-induced Sema3A cKO mice and Sema3A[fl/fl] mice. However, the ratio of macrophage number in the allergy group to that in the control group showed no difference between the two types of mice (Fig. 6b). Similarly, although the number of DCs increased in the Ni allergy-induced Sema3A cKO group compared to that in the Ni allergy-induced Sema3A[fl/fl] group, the ratio of DCs in the allergy group to that in the control group showed no difference between the two types of mice (Fig. 6c). Although Sema3A-specific deletion in keratinocytes caused upregulated immune cell infiltration, the unchanged ratio of CD4[+]/CD8[+] T cell number and the ratios of the number of macrophages or DCs in the allergy/control groups indicate that the deletion of Sema3A in keratinocytes would not significantly affect the development of Ni allergy.

**Sema3A induces a T helper (Th) 1 response rather than Th2 response.** To examine the effect of Sema3A on T cell differentiation, quantitative RT-PCR analysis of PBS/Ni/Concanavalin A (ConA)-stimulated splenic T cells in the absence or presence of recombinant Sema3A was performed. The results revealed that addition of Sema3A to Ni-stimulated splenic T cells significantly increased the expression levels of IL-2 and INF-γ, which are expressed by Th1 cells. However, IL-4 and IL-13, which are expressed by Th2 cells, did not show an increase in expression upon the addition of recombinant Sema3A protein (Fig. 7). This indicates that Sema3A induces a Th1 response, but not Th2 response, which may be the key to promoting the process of Ni allergy that is mainly mediated by Th1 response in Sema3A cKO mice.

## Discussion

Sema3A, a member of the semaphorin family, is known to function as an axonal repulsion factor through interaction with the plexinA/Nrp1 receptor[12]. The significance of Sema3A in regulating immune-mediated inflammation has been widely reported; for example, the expression level of Sema3A in the epidermis of patients with AD and psoriasis was remarkably decreased compared to that in healthy volunteers[28]; in the epidermis of patients with AD, epidermal hyperinnervation and high levels of nerve growth factor (NGF) were observed[29]. Since Sema3A is known to inhibit the intraepidermal extension of peripheral nerves and has been reported to abolish the growth-promoting effect of NGF on sensory afferents in the adult rat spinal cord and mouse embryonic neurons[30–32], the epidermal innervation in AD and psoriasis is likely to be regulated by a balance of NGF and Sema3A, and the decreased Sema3A expression in the epidermis of these patients could result from increased NGF levels. In our study, the expression of Sema3A was increased in the inflammatory site of the skin of metal allergy mouse models, suggesting that Sema3A may be involved in the pathogenesis of both diseases through different mechanisms.

Keratinocytes, which make up the majority of epidermal cells, are the first cells to have access to metal ions entering from the outside. Previous studies have demonstrated that activated keratinocytes can release certain proinflammatory cytokines such as IL-1β and TNF-α, TSLP, and chemokines in response to pathogens, injury, wound healing, as well as cancers[7,33–35]. In our study, when keratinocytes came in contact with Ni ions, they abundantly secreted Sema3A and TNF-α. TNF-α has been reported to promote tissue repair and enhance keratinocyte motility and attachment[36,37]. It also regulates innate immunity and inflammation by inducing chemokines that attract neutrophils, macrophages, and skin-specific memory T-cells[38]. Recent studies on the effect of Sema3A on the immune response have reported its association with the TNF-α and MAPK

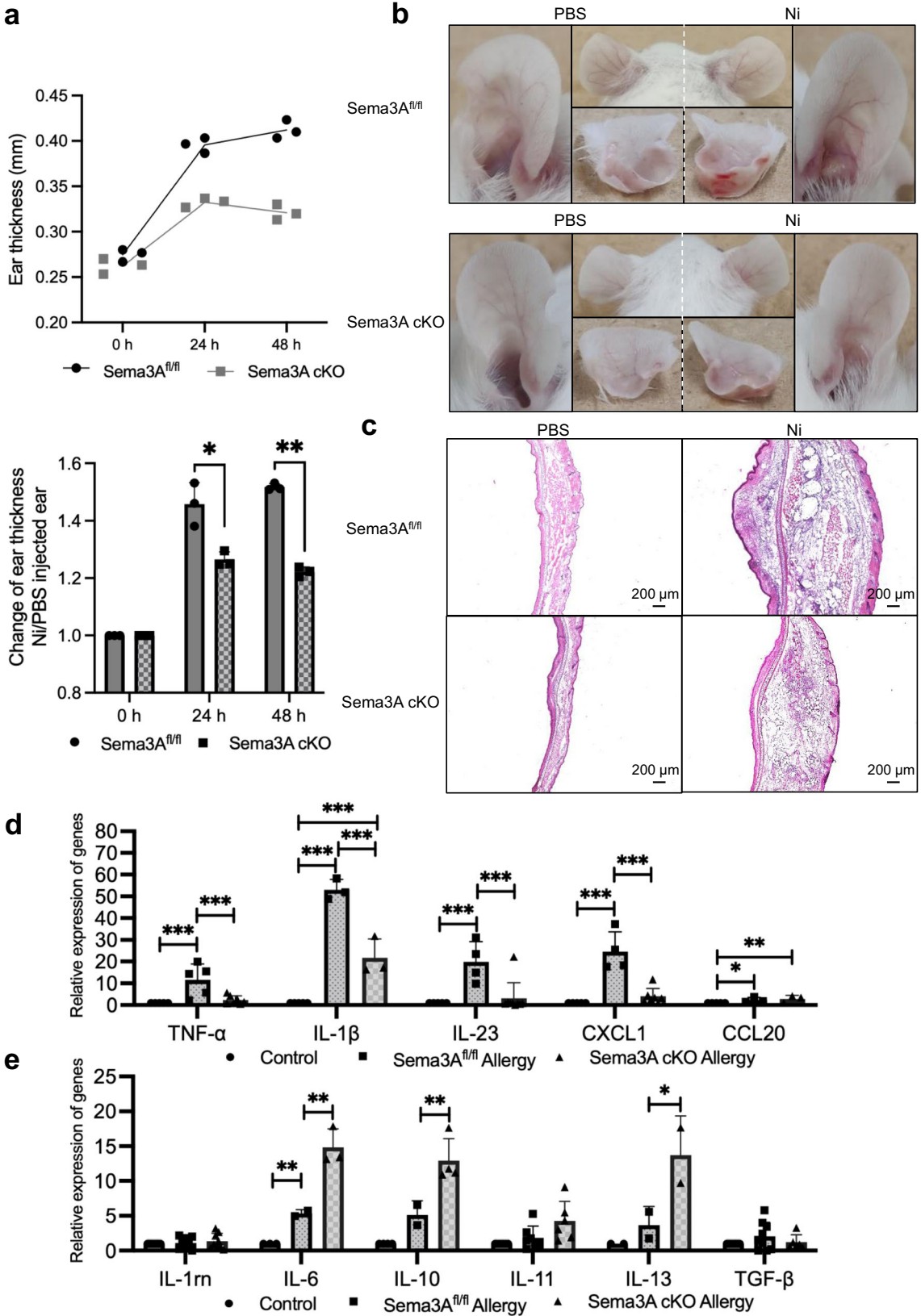

pathways[39]. In our study, both TNF-α and p38 kinase expression levels were inhibited when Sema3A was suppressed in keratinocytes. In previous studies, significantly decreased TNF-α expression in microglial cells in response to TLR4 stimulation was observed with the knockdown of the receptor of PlexinA1, which is a receptor of Sema3A[40,41]. These results reflect the possibility

that the production of TNF-α in keratinocytes stimulated with NiCl$_2$ depends on Sema3A secretion.

Meanwhile, keratinocytes stimulated with NiCl$_2$ showed activation of p38, and the inhibition of Sema3A with siRNA inhibited the activation of p38. It is known that p38 promotes the upregulation of IL-1β and TNF-α, which are strongly involved in

**Fig. 5 Allergy symptoms in Sema3A cKO mice are alleviated upon Ni re-challenge. a** DTH determined by measuring ear thickness after Ni re-challenge. The change in ear thickness in the two types of mice is shown in the upper panel. The ratio of allergy/control ear thickness of Sema3A[fl/fl] and Sema3A cKO mice is shown in the lower panel. Data are shown as mean ± SD of 3 mice/group. **b** Representative photos of the Ni allergy-induced ear and control ear of Sema3A[fl/fl] and Sema3A cKO groups. Photos show the ventral side (right and left), dorsal side (middle-upper), and cross-section (middle lower). **c** Histological images of the Ni allergy-induced mouse ear tissue and control ear tissue of Sema3A[fl/fl] and Sema3A cKO groups. Scale bar, 200 μm. The mRNA expression of related cytokines/chemokines (**d**) and anti-inflammatory cytokines (**e**) in Ni allergy-induced mouse ear tissue of Sema3A[fl/fl] ($N = 5$) and Sema3A cKO ($N = 9$) groups analyzed using quantitative RT-PCR 48 h after Ni re-challenge. Gene expression was normalized to levels of β-actin. Data are shown as mean ± SD and are representative of at least three independent experiments. *$p < 0.05$, **$p < 0.01$, ***$p < 0.001$. Two independent samples *t*-test for **a**, Two independent samples *t*-test and One-Way ANOVA with LSD for **d**, **e**. Supplementary Data and *p* values are provided as a Supplementary Data file.

---

skin inflammation[42,43]. In addition, studies using itch-deficient (Itch[−/−]) mice showed enhancement of phosphorylation of p38α in skin lesions, and the levels of proinflammatory cytokines such as TNF-α and IL-1β were increased in the skin of Itch[−/−] mice. Moreover, the inhibition of p38 significantly suppressed inflammation in the skin of Itch[−/−] mice[44]. Therefore, our results suggest that inhibition of Sema3A expression may have suppressed TNF-α secretion by inhibiting p38 in keratinocytes.

In our study, Sema3A cKO mice were protected against Ni allergy; although redness and cell infiltration were still observed in the ear tissue of Sema3A cKO mice, the conditions of severe edema and ear thickening were alleviated. As we did not observe a reduction in the number of T cells, macrophages, and DCs in the Ni allergy-induced Sema3A cKO mice, it is speculated that the alleviation of ear thickening was caused by decreased edema. Studies have reported that Sema3A could act as a vascular permeability factor and induce peripheral vasculature permeability[45,46]. Therefore, it is speculated that the reduced edema was caused by decreased vascular permeability due to Sema3A deletion.

We observed a decrease in the levels of proinflammatory factors and an increase in the levels of anti-inflammatory factors upon induction of Ni allergy in our study. IL-10 produced by Th2 cells targets macrophages and monocytes and inhibits the release of proinflammatory cytokines, including IL-β and TNF-α[47]. IL-13, reported to be primarily produced by activated Th2 cells, downregulates macrophage activity[48]. The significantly increased IL-10 and IL-13 levels in Sema3A cKO mice suggest an increased Th2 response due to the deletion of Sema3A in the epidermis. To verify this speculation, we cultured splenic T cells and stimulated them with NiCl₂ in the absence or presence of Sema3A protein. As DTH is reported to be mainly mediated by Th1 cells[49] and our results showed an increase in the levels of Th1-induced cytokines, but not Th2-induced cytokines, it is possible that Sema3A promotes the activation of Th1 cells to induce allergic reactions. Therefore, in the Sema3A cKO mice, the Th1 response is likely suppressed and the alteration in balance of the Th1/Th2 response might have led to an alleviated Ni allergy reaction.

Intradermal injection or ointment allocation of Sema3A were found to inhibit scratching behavior and alleviate AD-like symptoms compared to that in controls. Moreover, the nerve fiber number in the epidermis, inflammatory cell infiltration, and cytokine production were reduced in Sema3A-treated skin lesions[22,23]. Decreased Sema3A concentration may lead to the occurrence or enhancement of itching in patients with AD and NC/Nga mice[50,51]. In our study, we found that Sema3A can promote the differentiation of T cells to Th1 cells but not Th2 cells. Thus, in AD, which is a Th2-mediated disease, Sema3A induces T cell differentiation towards a Th1 response, thereby alleviating Th2 responses and reducing AD symptoms. In contrast, in DTH, which is a Th1-mediated disease, the deletion of Sema3A weakens the Th1 response and shifts the balance of Th1/Th2 response in the direction of Th2, thereby alleviating the allergic reactions.

In conclusion, through this study, we clarified that Sema3A could promote the development of Ni allergy by mediating the activation of MAPK and production of TNF-α. Deletion of Sema3A in keratinocytes does not influence the immune cell infiltration but reduces edema and ear swelling; it also hinders Th1 responses and results in alleviated Ni allergy. It is conceivable that Sema3A has the potential to be further developed to provide new ideas and therapeutic targets for metal allergy.

## Methods

**Mice.** Female C57BL/6J mice (8-week-old) were purchased from Charles River Laboratories, Inc. (Kanagawa, Japan). K5-cre[52] mice (STOCK-Tg(K5-Cre) Jt) were obtained from CARD R-BASE, Kumamoto University. K5-Cre-specificity was determined using forward and reverse primers GAACCTGATGGACATGTTCA GG/AGTGCGTTCGAACGCTAGAGCCTGT; and PCR conditions 95 °C for 5 min, [(95 °C for 30 s, 62 °C for 30 s, 72 °C for 30 s) × 35 cycles], then 72 °C for 5 min (band length 320 kb)[53]. Sema3A[fl/fl] mice (ICR.Cg-Sema3a < tm1.2Tyag > / TyagRbrc, RBRC01106) were provided by RIKEN BRC through the National BioResource Project of the MEXT/AMED, Japan. The genotyping of Sema3A[fl/fl]-specificity was performed determined using Sema3A's 5′ intron: ACAACGCT TGCCTCGGGAGGTAAA (presence of 1600bp band indicating floxed-type gene) and Sema3A's 3′ intron: ATGGTTCTGATAGGTGAGGCATGG (presence of 1200 bp band indicating wild-type gene); and PCR conditions 94 °C for 5 min, [(94 °C for 30 s, 60 °C for 30 s, 72 °C for 2 min) × 35cycles], then 72 °C for 5 min, according to the previous study[54]. Sema3A[fl/fl] conditional mice were crossed with K5-Cre mice to generate Sema3A cKO mice. Wild-type mice of the same genetic background strains were used as control animals. All mice used were aged 6-8 weeks. All mice were maintained under specific pathogen-free conditions and fed with autoclaved diet and water in the animal facilities at Tokushima University, they were treated in accordance with the National Institutes of Health Guide for the Care and Use of Laboratory Animals. All experimental procedures were approved by IACUC (No. T2019-51) and Institute for Genome Research (No.30-46) of Tokushima University.

**Reagents.** Nickel(II) Chloride was obtained from Wako Pure Chemical Corporation. Freund's incomplete adjuvant (IFA) and complete adjuvant (CFA) were from MP Biomedicals. The primary antibody specific for Sema3A was from Abcam. Antibodies specific for phospho-p38 (Thr180/Tyr182) and p38 were obtained from Cell Signaling Technology. Rabbit antibody to GAPDH was obtained from Osenses. Rabbit antibody to β-actin was obtained from Bioss. Horseradish peroxidase (HRP)-conjugated anti-rabbit secondary antibody were obtained from Cell Signaling. FITC anti-mouse CD11c, Alexa Fluor 488 anti-mouse CD11b, PE anti-mouse F4/80, PE anti-mouse I-A/I-E, FITC anti-mouse CD3, PE/Cyanine7 anti-mouse CD4 and PE anti-mouse CD8a were purchased from BioLegend. APC-Cy™7 Rat Anti-Mouse CD45 and Purified Rat Anti-Mouse CD16/CD32 (Mouse BD Fc Block™) were from BD Biosciences. 7-AAD Viability Staining Solution was from Invitrogen. Anti-rabbit Alexa Fluor 488 secondary antibody was from Abcam. Readidrop™ Propidium Iodide was from Bio-Rad Laboratories. The recombinant mouse Sema3A were purchased from R&D Systems.

**Induction of Ni allergy.** Ni allergy was induced on mouse ears according to our previous studies[9–11]. In brief, 25 μl 1 mM NiCl₂ with 25 μl IFA was intraperitoneally injected to mice for initial immunization. After two weeks, 10 μl 0.2 mM NiCl₂ with 10 μl CFA was intradermally injected into the ear skin as a second challenge. For negative control, PBS with CFA was injected. 48 h later, DTH reaction was observed and confirmed by measuring the ear thickness of mice.

**Histology and immunofluorescence microscopy.** Frozen sections (4–8 μm) were stained with H&E. Sections for immunofluorescence staining were fixed with pre-cooled acetone (4 °C) for 10 minutes. After this, the slides were rinsed with PBS and

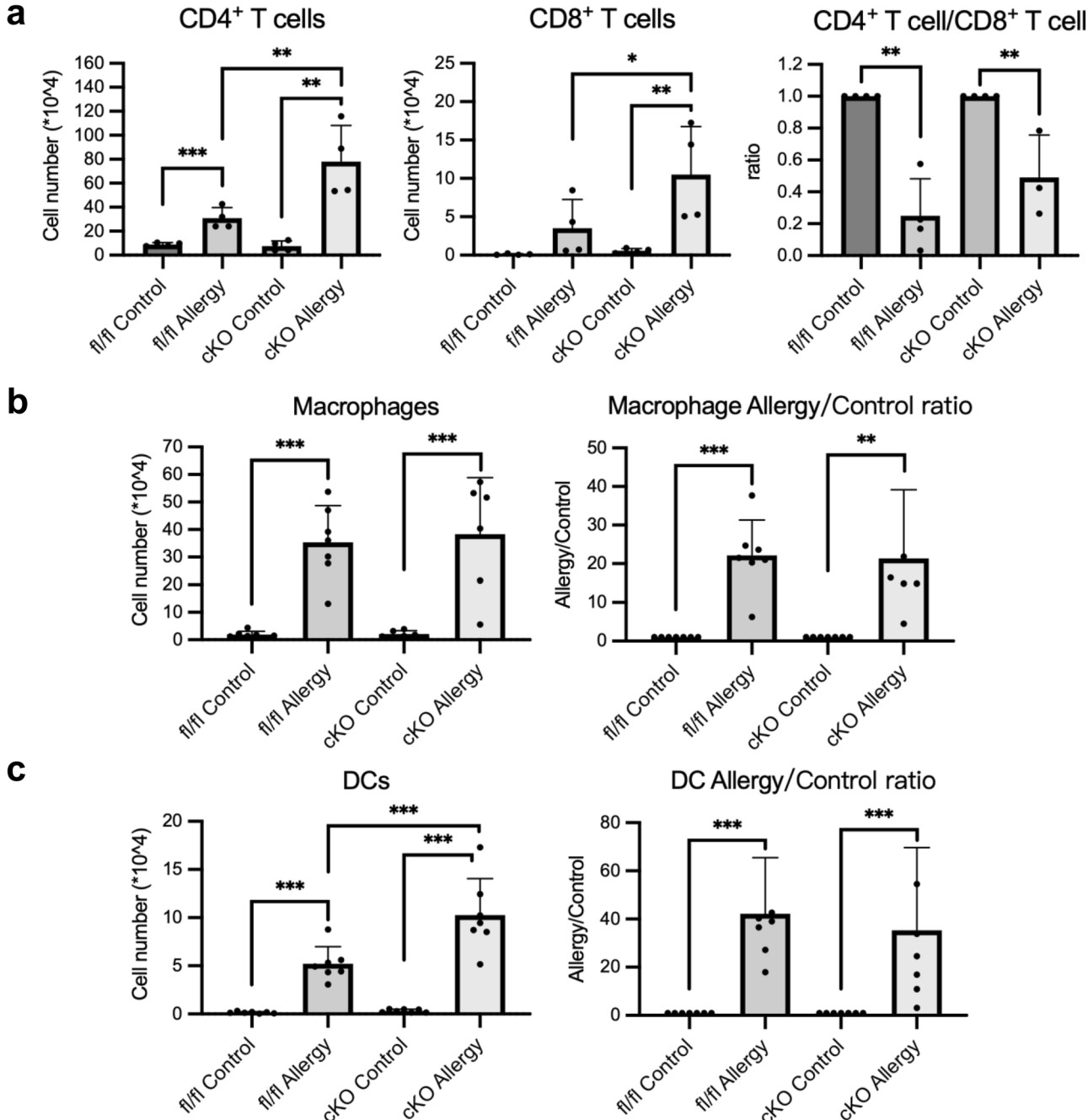

**Fig. 6 The number of infiltrated immune cells is altered upon Sema3A deletion. a** The populations of CD4+ T cells (left panel) and CD8+ T cells (middle panel) as well as the CD4+ T cell/CD8+ T cell ratio (right panel) are shown. Data are shown as mean ± SD of 4 mice/group. **b** The populations of macrophages (left panel) as well as the macrophage Allergy/Control ratio (right panel) are shown. Data are shown as mean ± SD of 7 mice/group. **c** The populations of DCs (left panel) as well as the DC Allergy/Control ratio (right panel) are shown. Data are shown as mean ± SD of 7 mice/group. Data are shown as mean ± SD and are representative of at least three independent experiments. $*p < 0.5$, $**p < 0.01$, $***p < 0.001$. Two independent samples $t$-test and One-Way ANOVA with LSD. Supplementary Data and $p$ values are provided as a Supplementary Data file.

blocked with 5% goat serum for 30 min. A 1:300 dilution of primary antibodies or fluorescently-conjugated antibodies was applied to the sections and incubated in a dark humidified chamber overnight at 4 °C, subsequently followed by incubation with Alexa Fluor secondary antibody at room temperature for 1 h for not fluorescently-conjugated antibodies. The nuclei were stained with DAPI solution (0.1 µg/ml; Cell Signaling Technology, Danvers, MA, USA). Immunofluorescence images were captured using Keyence all-in-one fluorescence-microscope BZ-X800 (magnification, ×200; ×400; ×1000; KEYENCE CORPORATION, Tokyo, Japan). BZ-X800 Analyzer software was adapted for the extraction and quantification of the imaging data.

**Epidermal sheet preparation**. Epidermal sheets were prepared using a modification of previously-reported method[55,56]. In brief, the ears were peeled into

halves and put dermis-side down floating on a dish of cold ammonium thiocyanate. After incubated 13 min at 37 °C, the epidermal sheets were peeled with small tweezers. The epidermis flats were washed with PBS and fixed with cold acetone, then used for the following immunofluorescence staining like frozen sections.

**Flow cytometry**. The single-cell suspensions of mouse ears were prepared based on the previous method[57,58]. Briefly, the ears were split and cut, incubated with a solution of RPMI containing 1 mg/ml DNase I (NIPPON GENE, Tokyo, Japan) and 1 mg/ml collagenase (Worthington Biochemical, Lakewood, NJ, USA) 90 min at 37 °C. $1 × 10^6$ cells of homogeneous cell suspension were incubated with Purified Rat Anti-Mouse CD16/CD32 for 5 min and stained with FITC anti-mouse CD11c, PE anti-mouse F4/80, APC-Cy™7 anti-mouse CD45, FITC anti-mouse CD3, PE/

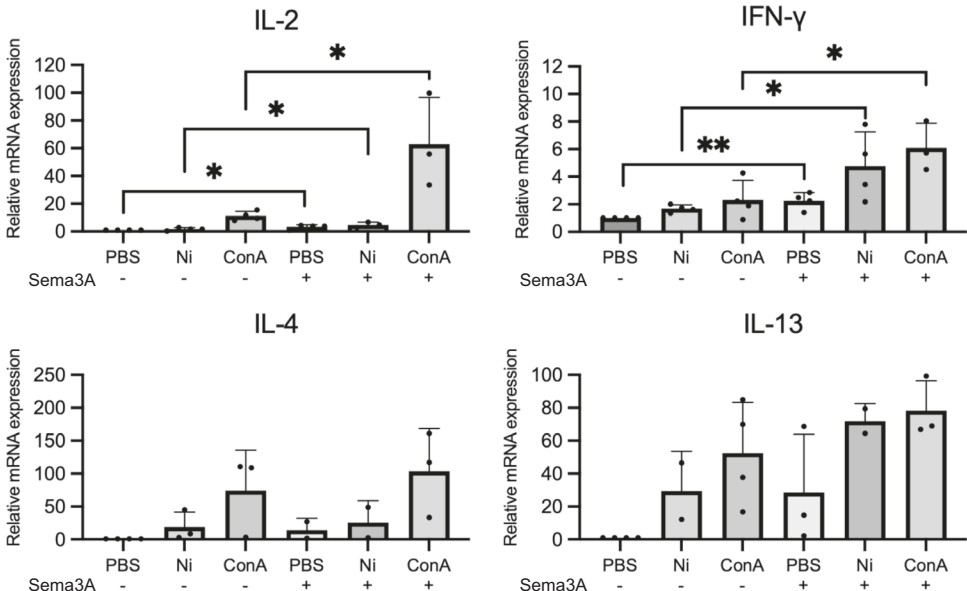

**Fig. 7 Sema3A induces T cell differentiation.** The mRNA expression of Th1-induced (IL-2 and IFN-γ) and Th2-induced (IL-4 and IL-13) cytokines in PBS/Ni/ConA-stimulated splenic T cells in the absence or presence of recombinant Sema3A analyzed using quantitative RT-PCR. Gene expression was normalized to levels of β-actin. $N = 4$. Data are shown as mean ± SD and are representative of four independent experiments. *$p < 0.5$, **$p < 0.01$. Two independent samples $t$-test. Supplementary Data and $p$ values are provided as a Supplementary Data file.

Cyanine7 anti-mouse CD4, PE anti-mouse CD8a, 7-AAD according to the recommended concentration. FITC Armenian Hamster IgG and PE Rat IgG2a, κ were used for isotype control. $1 \times 10^6$ Pam2.12 cells were stained with Readidrop™ Propidium Iodide (Bio-Rad Laboratories, Inc., CA, USA) according to the manufacturer instructions. Cells were analyzed on BD FACSVerse™ (BDBiosciences, San Jose, CA, USA).

**Cell culture and siRNA transfection.** Mouse keratinocyte cell line Pam2.12 was kindly provided by Dr. S. H. Yuspa (National Cancer Institute, Bethesda)[59]. $1 \times 10^7$ cells/10 cm dish were cultured in DMEM (nacalai tesque, Kyoto, Japan) supplemented with 10% FBS and 1% penicillin/streptomycin/amphotericin B in a humidified atmosphere of 5% $CO_2$ at 37 °C. Pam2.12 cells were stimulated with 250 μM $NiCl_2$ for 0, 6, 12, 24, 48, 72 h before analysis (The concentration was decided according to previous study[11]). The Sema3A esiRNA was obtained from Sigma-Aldrich (Tokyo, Japan). Pam2.12 cells were transfected with 50 nM of Sema3A siRNA using INTERFERin (Polyplus transfection, Illkirch, France) in half-decreased-volume serum-free DMEM, after 4 h, complete DMEM was added to restore the usual culture volume. Medium was changed 24 h after transfection and 250 μM $NiCl_2$ was added to the cells for another 24 and 48 h.

T cells were collected from mouse spleen. The spleen was removed and grinded with the flat end of a syringe to pass through a cell strainer into a 50 ml tube. After washed with 5 ml of PBS twice, the cell pellet was re-suspended with red blood lysis buffer for 15 minutes. Cells were washed and spined down, cultured in a 6-well plate with 2 ml of RPMI containing 10% FBS and 1% penicillin/streptomycin/amphotericin B per well for 24 h. 10 μl of PBS, 50 μM of $NiCl_2$, 5 μg/ml of ConA was added to cells with or without recombinant mouse Sema3A (100 ng/ml) for 48 h. After which total RNA of cells were collected for subsequent analysis of quantitative RT-PCR.

**Immunocytochemistry.** Pam2.12 cells were seeded in dish with glass coverslips in the bottom. 250 μM $NiCl_2$ was added when cells grew to 40% confluency. After $NiCl_2$ stimulation for 24 or 48 h, coverslips were removed and washed with PBS, followed by fixation with 4% paraformaldehyde at room temperature for 10 min. Then the coverslips were incubated in 0.25% Triton X-100 in PBS at room temperature for another 10 min and blocked in 3% BSA for 30 min. Finally, they were stained with immunofluorescence like frozen sections.

**Quantitative RT-PCR.** The total RNA of Pam2.12, mouse ears and splenic T cells were extracted. cDNA was made using PrimeScript™ RT reagent Kit (TaKaRa Biotech, Shiga, Japan). TaKaRa Ex Taq® DNA Polymerase (TaKaRa Biotech, Shiga, Japan) was used for preliminary experiment to confirm expressions. qPCR was performed using TB Green® Premix Ex Taq™ II (TaKaRa Biotech, Shiga, Japan) in a total volume of 15 μl with ABI7300 Real-time PCR System (Applied Biosystems, MA, USA) according to manufacturer instructions. The polymerase was initially activated at 95 °C for 30 s, then the PCR was run in 40 cycles of denaturation at 95 °C for 5 s and annealing/extension at 60 °C for 31 s. The sequences for primers

were as follows: *Sema3a*, forward: 5′-GAAGAGCCCTTATGATCCCAAAC-3′; reverse: 5′-AGATAGCGAAGTCCCGTCCC-3′. *Tnf-alpha*, forward: 5′-ATGGCCTCCCTCTCATCAGT-3′; reverse: 5′-CTTGGTGGTTTGCTACGACG-3′. *Il1b*, forward: 5′-GCTGAAAGCTCTCCACCTCA-3′; reverse: 5′-AGGCCACAGGTATTTTGTCG-3′. *Il23*, forward: 5′-CCAGCAGCTCTCTCGGAATC-3′; reverse: 5′-TCATATGTCCCGCTGGTGC-3′. *Cxcl1*, forward: 5′-CCGAAGTCATAGCCACACTCAA-3′; reverse: 5′-GCAGTCTGTCTTCTTTCTCCGTTA-3′. *Ccl20*, forward: 5′-GTACTGCTGGCTCACCTCTG-3′; reverse: 5′-CTTCATCGGCCATCTGTCTTGTG-3′. *Il1rn*, forward: 5′-TGTGCCTGTCTTGTGCCAAGTC-3′; reverse: 5′-GCCTTTCTCAGAGCGGATGAAG-3′. *Il6*, forward: 5′-TACCACTTCACAAGTCGGAGGC-3′; reverse: 5′-CTGCAAGTGCATCATCGTTGTTC-3′. *Il10*, forward: 5′-CGGGAAGACAATAACTGCACCC-3′; reverse: 5′-CGGTTAGCAGTATGTTGTCCAGC-3′. *Il11*, forward: 5′-CTGACGGAGATCACAGTCTGGA-3′; reverse: 5′-GGACATCAAGTCTACTCGAAGCC-3′. *Il13*, forward: 5′-AACGGCAGCATGGTATGGAGTG-3′; reverse: 5′-TGGGTCCTGTAGATGGCATTGC-3′. *Tgf-beta*, forward: 5′-TGATACGCCTGAGTGGCTGTCT-3′; reverse: 5′-CACAAGAGCAGTGAGCGCTGAA-3′. *Il2*, forward: 5′-GCTGTTGATGGACCTACAGGA-3′; reverse: 5′-TTCAATTCTGTGGCCTGCTT-3′. *Ifn-gamma*, forward: 5′-GGCCATCAGCAACAACATAAGCGT-3′; reverse: TGGGTTGTTGACCTCAAACTTGGC-3′. *Il4*, forward: 5′-TGTACCAGGAGCCATATCCAC-3′; reverse: 5′-GTTCTTCGTTGCTGTGAGGAC-3′. *Actb*, forward: 5′-TCTGGCTCCTAGCACCATGAAGA-3′; reverse: 5′-GGGACTCATCGTACTCCTGCTTG-3′. Gene expression was normalized to levels of β-actin.

**Western blotting analysis.** $NiCl_2$-stimulated Pam2.12 cells were washed in cold PBS before lysis in 60 μl of RIPA buffer (FUJIFILM Wako Pure Chemical Corporation, Osaka, Japan) containing 1:100 EZBlock™ Protease Inhibitor Cocktail (BioVision, San Francisco, CA, USA). Tissue extracts were prepared by homogenizing mouse ear tissue in lysis buffer. Proteins were separated in a 10% SDS-PAGE acrylamide gel for Pam2.12, transferred onto a polyvinylidene fluoride membrane. Membranes were incubated with primary antibodies against Sema3A, phosohorylated-p38 MAPK at 4 °C overnight. Total p38 MAPK and β-actin or GAPDH were incubated at room temperature for 1 h. All the primary antibodies were used at a dilution 1:1000. After washing with TBST (0.05 M Tris-HCl, 0.15 M NaCl, pH7.6), a 1:20000 dilution of HRP-conjugated anti-mouse secondary antibody was applied to the membrane for 1 h at room temperature. Membranes were detected with ECL Prime reagents (GE healthcare, Chicago, IL, USA) using ChemiDoc XRS (Bio-Rad, Hercules, CA, USA).

**ELISA.** Pam2.12 cells were seeded in 96-well dish and stimulated with 250 μM $NiCl_2$ for 0, 12, 24, 48, 72 h before analysis of Sema3A production in medium using a mouse Sema3A ELISA kit (Signalway Antibody, College Park, MD, USA). Pam2.12 cells seeded in 96-well dish was performed with Sema3A siRNA transfection using INTERFERin as described above to suppress Sema3A expression. Culture supernatant 24 h after $NiCl_2$-stimulation was collected and stored at −20 °C until analysis. The TNF-α production was measured using a mouse TNF-α

ELISA (eBioscience, San Diego, CA, USA), according to the manufacturer instructions.

**Statistics and reproducibility**. All experiments were repeated at least three times with similar results. For the Ni allergy induction on mouse ears, at least 3 mice were used for each group of C57BL/6J, Sema3A^fl/fl, and Sema3A cKO. Experimental values are given as means ± SD. The statistical difference was determined by One-Way ANOVA with LSD (Least Significant Difference), Two independent samples $t$-test. Statistical significance is presented in the following manner: $*p < 0.05$, $**p < 0.01$, $***p < 0.001$.

**Reporting summary**. Further information on research design is available in the Nature Research Reporting Summary linked to this article.

## Data availability

The authors declare that the data supporting the findings of this study are available within the paper and from the authors upon reasonable requests. Supplementary Data files are provided with this paper.

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

## Acknowledgements

This study was supported by the Support Center for Advanced Medical Science, Tokushima University Graduate School of Biomedical Sciences. This work was supported by JSPS KAKENHI Grant Number JP18K09661. L.L. would like to thank the Otsuka Toshimi Scholarship Foundation for its support.

## Author contributions
L.L., M.W. and T.I. conceived and designed the study. L.L., M.W., N.M. and M.F.Y. performed the in vitro experiments and acquired the data. L.L. and M.W. performed the animal experiment and obtained the data. L.L. and M.W. interpreted and analyzed the all data, and wrote the manuscript. All authors reviewed the results and provided feedback on the manuscript.

## Competing interests
The authors declare no competing interests.
