## [Peer Review File · Communications Biology]

Reviewers' comments:

Reviewer #1 (Remarks to the Author):

In their manuscript "Semaphorin 3A promotes the development of nickel allergy" the authors aim to elucidate the pathogenic mechanism of nickel allergy and invoke a role for Sema3A. The authors use a previously established mouse Ni allergy model in the current study. Using this model, the authors had previously shown a role for MAPK and TSLP as well as Sema7A in the development of Ni allergy.

The study is well conducted, with rigorous scientific approach and data analysis. The study adds to the literature around the role of semaphorins in a variety of immune-mediated conditions and specifically expands on the pathogenic mechanisms of metal allergy. The authors use a variety of experimental approaches to explore the role of Sema3A in their model.

Although well conducted, one point that is somewhat nebulous and that merits more explicit explanation is why Sema3A was chosen as the protein to study. Are there any clinical or translational exploratory data suggesting a role for Sema3A in metal allergy? The authors hint at this in the discussion on page 12 and perhaps this idea can be incorporated into the introduction for a more linear story line. Additionally, some of the conclusions are not fully supported by the data presented. Overall a meritorious study that I believe will be strengthened by the comments below.

Figure 1:

Can the authors clarify why the standard errors are the same for all values in their statistical analysis tables?

Figure 1b: the authors should include imaging performed on cells at 72h. The Western blot and ELISA data suggest that the Sema3A is cell associated first and then released. Imaging data would support this.

Figure 2:

What is the knockdown efficiency? Can the authors provide data to show this?

Figure 3:

Can the authors clarify where the ear thickness measurements are taken?

In 3d the authors state that the negative control is primary without secondary, is this correct? Is the green fluorescence just background? Also, is the isotype control staining from a control or a Ni-challenged ear?

Again in 3d following Ni stimulation, it appears that the Sema3A expression (if the green fluorescence is indeed Sema3A) is now mostly in the epidermis and not so much in the dermal layer. Is this the case? How do the authors explain this? What cell types are expressing Sema3A under control vs Ni-treated conditions?

Figure 4:

The authors argue that there is an increase in the number of different immune cells, but numbers are not actually reported. Providing percentages of cells is NOT equivalent to absolute numbers. The authors should provide absolute cell numbers.

Additionally, the authors need to provide a clear gating strategy for how cell populations are identified as well as clone and dilution data for the antibodies used. This is now standard reporting in most scientific papers.

There is clearly an increase in the fraction of CD11b+ cells. What are these cells? Is there a neutrophilic infiltrate with this model?

CD11b+CD115+ cells: It is very difficult to argue that the differences between control and allergy are significant and it seems that the main driver of the significant differences is the 3rd experiment, based on the raw data. Additionally, this reviewer wonders about a possible compensation issue, given the shape of the curves. Also, in the associated imaging data, it's difficult to argue that there is an increase in CD115+ cells. Perhaps a redistribution, not an increase.

The CD11c+ cells seem to be located in a compartment that is structurally distinct from the

DC11b+ cells (the DAPI stain does clearly delineate an epidermal layer in 4c and 4f, but not in 4i). Where are the CD11c+ cells?

Are the authors able to provide a quantitative analysis of the imaging data?

Did the authors evaluate T cells in this model? In a DTH model, DC4 T cells are likely to be implicated.

Figure 5:

Can the authors provide the actual ear thickness data for the *Sema3A* cKO under the experimental conditions outlined in 5b like they do for 3e?

Figure 6:

The authors again argue that numbers of cells are changing when comparing experimental groups but do not provide any absolute count data. Reporting fractions or percentages of cells is very different than reporting absolute numbers. For example, 1% of 100 cells is the same absolute number as 0.1% of 1000 cells. Then one must determine what the contribution of these or other cells is to the immunological mixture in the particular tissue compartment analyzed. Please provide absolute numbers.

Again, the F4/80/CD206 data seem to be driven by one animal per group (allergy 4 in *Sema3A* fl/fl and Allergy 3 in *Sema3A* cKO). Can the authors revisit the analysis of these data?

The authors suggest that there is an increase in the M2 macrophage population compared to the M1 population and that this provides an anti-inflammatory environment. Did the authors measure anti-inflammatory cytokine levels? Additionally, what happens to TSLP in the knockout model?

Additionally, what changes take place with respect to infiltrating immune cell populations? Can the authors provide data similar to what is provided in Figure 4 for the WT animals? In the discussion the authors claim that *Sema3A* leads to CXCL1 expression and that this is responsible for immune cell trafficking to the ear. In the cKO experiments, the authors do not show how the immune cell composition in the ears changes. They explore macrophage populations but do not expand on other cell types. These data would strengthen their argument.

Discussion:

The sentence at lines 204-207 is unclear. Please revise.

Minor comments:

- Figure 1 legend has SMA3A instead of SEMA3A. Please correct the typo.
- Also, I suggest changing the figure titles to statements to reflect the results described.
- The statistical comparison scheme in Figure 1c and elsewhere (a, b, c, etc) is not immediately intuitive and should be described in the figure legends and not be only available in the source data files.
- Control GAPDH bands for some of the blots described in Figure 1c are suboptimal and likely difficult to interpret. Suggest choosing another protein to normalize against.
- Also, why is *Sema3A* in some blots represented by two bands?

Reviewer #2 (Remarks to the Author):

In the manuscript 'Semaphorin3A Promotes the Development of Nickel Allergy', the authors investigate the role of *Sema3A* in the immune response to nickel. They show that nickel stimulation induces an up-regulation of *Sema3A* on the keratinocyte cell-line PAM2.12 in a time-dependent manner. In correlation, they show that *Sema3A* is upregulated in the skin by using an in vivo model for nickel allergy. Furthermore, they show that *Sema3A* cKO mice have a reduced response to nickel compared to WT mice. Finally, they show an increased M2/M1 ratio in the ear skin of *Sema3A* cKO mice compared to WT mice during the response to nickel. These are interesting findings, but the following additional points need to be addressed.

Major points:

- 1) In figure 2 the authors use siRNA to knock down the nickel-induced upregulation of Sema3A on PAM2.12. Using this setup, the author finds reduced TNF α production and p38 phosphorylation induced by nickel stimulation. However, they do not show the mechanism behind this – does Sema3A work in an autocrine manner on the KC? Please show the receptor expression on the KC (in general, it would be nice if the authors included the receptor expression on different cellular subsets found in the skin during the response to nickel). To further confirm that Sema3A can stimulate the KC, please stimulate the KC with either soluble Sema3A or do blocking experiments.
- 2) In figure 3 the author shows that IL-17, IL-23, CCL20 and CXCL1 are all up-regulated in their nickel-model. In figure 5 the author shows that IL-1 β , TNF α , IL-23 and CXCL1 are all reduced in Sema3A cKO mice compared to WT mice during the response to nickel. Please show all cytokine/chemokines in both figure 3 and 5.
- 3) Nickel allergy is a T cell mediated disease and Sema3A is known to play a role in T cell activation/function. Still, the authors focus on the effect of Sema3A on the macrophage phenotype. Please include data showing the effect of Sema3A on the T cell response induced by nickel.

Minor point:

The manuscript needs to be carefully checked for spelling mistakes and the authors need to ensure that it is written in a correct English.

Reviewer #3 (Remarks to the Author):

General comments:

The manuscript described the involvement of Sema3A on the Ni allergy. Various studies already reported that Sema3A inhibit scratching behavior in atopic dermatitis. The manuscript is novelty that the blocked endogenous Sema3A reduced the Ni allergy. However, the study is the unreliability of the experiments results, the study is not recommended for publication in Communication Biology.

Major comments:

1. They compared the between PBS and Ni challenge in sensitized mice. These data do not exclude the possibility of toxicity from Nickel. If you clarify the involvement of Sema3A in Ni allergy, not toxicity, you should compare the between sensitized and unsensitized mice.
2. They should clarify the Sema3A effect at the sensitization phase or effector phase.
3. They clarified that Ni-induced Sema3A mediates TNF production and MAPK activation in vitro. But, in vivo data are insufficient to prove the Sema3A activates TNF and MAPK in Ni allergy.
4. In vitro data, they explain that the nickel concentration (250uM) was determined from reference 11. However, in reference 11, they used 10uM Ni because of over 100uM affect cell viability. They should explain why they use 250uM Ni.
5. In the previous data, they clarified the involvement of TSLP in Ni allergy (reference 10). You should state the effect of Ni-induced Sema3A on TSLP.
6. In flow cytometry data, the facs plot is unreliability, they should be reconsidered the experimental methods or add the isotype control.

Material and Method:

1. They should add the gating strategy in flow cytometry.
2. They should add the Fc block and dead cell staining in flow cytometry.
3. They should add the cell numbers of Pam212.

Result:

1. In Figure 1C, they should show adequate statistical analysis.
2. In Figure 1, They should add the data of cell viability and explain the Sema3A production is due to cell death or not.
3. In Figure 2A, they should state the time in control and Mock.
4. In Figure 2A, why they compare between control-control to mock-nickel?
5. In Figure 3C, the histological image show the ear thickness of Ni is 1.2mm, but in Figure 3E, the

ear thickness of Ni is only 0.45mm. Why does it make such a difference?

6. In Figure 3D, Isotype control are too dyed and should take adequate control.

7. In Figure 3, they determined the involvement of Ni-induced Sema3A to TNF and MAPK activation. Why they do not check the these ?

8. In Figure 4 A,D,G, macrophages in the ear are a well-separated cell population, the staining and experimental methods should be reconsidered.

9. In Figure 4 B,E,H, they should show the cell numbers.

10. In Figure 5B, they should show the data by ear thickness (mm).

11. In Figure 6, they should state the time after Ni challenge.

12. In Figure 6A, they defined M1and M2 macrophage by F4/80 and CD206. This is not sufficient, and the cells should be firmly stained with the M1 marker.

13. In Figure 6A, M1/M2 population is 4.18%/1.94% in Sema3A fl/fl mice and 4.8% / 2.54% in Sema3A cKO mice. Is this the correct way to present the data in Figure 6C?

Figure legend:

1. They should add the sample size.

2. In figure 1A, they need to modify Sma3A to Sema3A.

Point-by-point responses to reviewer comments:

Reviewer #1:

Before responding to the comments and requests, we would like to first thank you for all your helpful suggestions, which have greatly benefitted the revised manuscript. Our responses to each of your concerns are listed below; we hope that you find them satisfactory. All changes made to the manuscript are indicated using red font in the manuscript file.

1. (Introduction & Discussion) Why Sema3A was chosen as the protein to study. Are there any clinical or translational exploratory data suggesting a role for Sema3A in metal allergy?

As we have mentioned in the Introduction, Sema3A is a secretory member of the semaphorin family of proteins, which might function like cytokines or chemokines. Furthermore, it has been reported to play a crucial role in both intercellular and intracellular communication during various immune diseases. Recently, Sema3A was reported to alleviate the skin lesions and scratching behavior in an atopic dermatitis mouse model by interrupting the itch-scratch cycle. Considering the similar skin conditions between metal allergy-induced contact dermatitis and atopic dermatitis, we decided to study the role of Sema3A in metal allergy, which has not been reported yet. In addition, in our previous study, we showed the important role of Sema7A in the effector phase of metal allergy, and we hoped that through studying the role of Sema3A in metal allergy, a new potential target for further investigation and treatment could be developed.

2. (Figure 1) Can the authors clarify why the standard errors are the same for all values in their statistical analysis tables?

In the data analysis of SPSS, the standard error of each sample (raw data) is obtained by calculation in **descriptive statistics**. The standard error obtained by statistical calculation (such as variance analysis, etc.) in the **linear model** is not the standard error of the original data, but is calculated by considering several data processed at the same time as one sample. So several raw data processed at the same time will have the same standard error.

Standard error of the mean is calculated by taking the standard deviation and dividing it by the square root of the sample size.

$$\text{standard deviation } \sigma = \sqrt{\frac{\sum_{i=1}^n (x_i - \bar{x})^2}{n - 1}}$$
$$\text{standard error } (\sigma_{\bar{x}}) = \frac{\sigma}{\sqrt{n}}$$

Where:

\bar{x} = the sample's mean

n = the sample size

In the linear model where the raw data are processed as one sample, the standard deviation and the sample size would be the same, therefore, the same standard errors would be obtained.

3. (Figure 1b) The authors should include imaging performed on cells at 72h. The Western blot and ELISA data suggest that the Sema3A is cell associated first and then released. Imaging data would support this.

We have added the immunocytochemistry imaging of Sema3A expressed in Pam2.12 cells at 72 h post Ni stimulation. For consistency, we replaced images of other time points, which were taken in the same experimental condition with that of the 72 h

group. The corresponding results and figure legend (Fig. 1d) have been revised in the manuscript.

4. (Figure 2) What is the knockdown efficiency? Can the authors provide data to show this?

We have performed qPCR for the control group and Sema3A siRNA-transfected group. The relative Sema3A expression in these two groups is shown in the table below. The knockdown efficiency of Sema3A siRNA was **57.00%** by calculation.

Pam2.12	Control	Si
Sample1	1	0.462813
Sample2	1	0.39587
Sample3	1	0.43146
MEAN	1	0.430048
SD	0	0.033493

5. (Figure 3) Can the authors clarify where the ear thickness measurements are taken?

As shown in the picture below, the white spot in the center of the soft area of the mouse ear is where we injected and took the measurement of ear thickness.

6. (Figure 3) In 3d the authors state that the negative control is primary without secondary, is this correct? Is the green fluorescence just background? Also, is the isotype control staining from a control or a Ni-challenged ear?

We apologize for the mistake in our description. The negative control included staining with only secondary antibodies without the primary antibody. We have revised this part in the manuscript as: “Staining with only anti-rabbit Fluor Alexa 488 secondary antibody was set as the isotype control”.

As shown in the following image at a lower magnification (200×), the cartilage (Δ), muscle, and fatty tissue (\circ) would stain green as the background.

The isotype control image in the manuscript was from a control ear tissue and here we have shown an isotype control image from the Ni-challenged ear tissue (below).

(isotype control image from control ear)

(isotype control image from Ni-challenged ear)

7. (Figure 3) Again in 3d following Ni stimulation, it appears that the Sema3A expression (if the green fluorescence is indeed Sema3A) is now mostly in the epidermis and not so much in the dermal layer. Is this the case? How do the authors explain this? What cell types are expressing Sema3A under control vs Ni-treated conditions?

Sema3A is reported to be expressed on neuronal cells, glandular epithelial cells, and endocrine cells. In our study, Sema3A was mainly expressed by keratinocytes in the epidermis, which is similar to that reported in other studies (kindly see the references below). Keratinocytes basically express Sema3A but show increased secretion upon Ni stimulation. In addition, we observed Sema3A expression in the dermis, which might have spread after secretion from keratinocytes or combined with receptors on other immune cells such as DCs.

- Kurschat, P., Bielenberg, D., Rossignol-Talandier, M., Stahl, A., Klagsbrun, M. Neuron restrictive silencer factor NRSF/REST is a transcriptional repressor of neuropilin-1 and diminishes the ability of semaphorin 3A to inhibit keratinocyte migration. *J. Biol. Chem.* **281**, 2721-2729 (2006). doi: 10.1074/jbc.M507860200. Epub 2005 Dec 5. PMID: 16330548.
- Tominaga, M., Ogawa, H., Takamori, K. Decreased production of semaphorin 3A in the lesional skin of atopic dermatitis. *Br. J. Dermatol.* **158**, 842-844 (2008). doi: 10.1111/j.1365-2133.2007.08410.x. Epub 2008 Jan 30. PMID: 18241279.

8. (Figure 4) The authors should provide absolute cell numbers of different immune cells. Additionally, the authors need to provide a clear gating strategy for how cell populations are identified as well as clone and dilution data for the antibodies used.

We have modified and described the gating strategy and provided absolute cell numbers of the different immune cells in Figure 4 in the revised manuscript.

9. (Figure 4) There is clearly an increase in the fraction of CD11b⁺ cells. What are these cells? Is there a neutrophilic infiltrate with this model?

We considered the increased CD11b⁺ cells in our study to be monocytes and macrophages, although we did not check for the neutrophil marker by flow cytometry.

In our H&E staining results, we found an increase in neutrophil infiltration in Ni allergy-induced mouse ear tissue, as shown in the following pictures (arrow heads).

Control (×400 magnification)

Allergy ($\times 400$ magnification)

10. (Figure 4) CD11b+CD115+ cells: It is very difficult to argue that the differences between control and allergy are significant and it seems that the main driver of the significant differences is the 3rd experiment, based on the raw data. Additionally, this reviewer wonders about a possible compensation issue, given the shape of the curves. Also, in the associated imaging data, it's difficult to argue that there is an increase in CD115+ cells. Perhaps a redistribution, not an increase.

According to your suggestion, we have modified the gating strategy and calculated the absolute number of each type of cell in the whole ear tissue. The data are shown in Figure 4 and the Source Data file, and the corresponding text has been revised in the manuscript.

11. (Figure 4) The CD11c+ cells seem to be located in a compartment that is structurally distinct from the DC11b+ cells (the DAPI stain does clearly delineate an epidermal layer in 4c and 4f, but not in 4i). Where are the CD11c+ cells?

Figures 4c and 4f (now Fig. 4g) are images of horizontally frozen sections of mouse ear tissues that showed a clear structure of the epidermal layer. However, Figure 4i (now Fig. 4h) shows images of the epidermal sheet. We peeled off the epidermis part of the ear tissues and stained them as mentioned in the Materials and Methods. The CD11c+ cells in Figure 4i (now Fig. 4h) show the sites of DCs (mostly Langerhans cells) in the epidermis.

12. (Figure 4) Are the authors able to provide a quantitative analysis of the imaging data?

We have performed a quantitative analysis using the BZ-X800 Analyzer software. The analysis results of the imaging data shown in Figure 4 as well as in Figures 1 and 3 are attached, and the analysis process and data are attached as a supplementary figure and Source Data file, respectively. In order to better arrange the figures, the panel sequence has been rearranged and the related description in the manuscript has

been revised.

13. (Figure 4) Did the authors evaluate T cells in this model? In a DTH model, DC4 T cells are likely to be implicated.

We have checked the T cells in this model and the result is shown below. The following figure shows the flow cytometry gating strategy for CD4⁺ T cells and CD8⁺ T cells.

As shown in the figure below, the number of CD4⁺ T cells and CD8⁺ T cells increased significantly in the Ni allergy-induced mouse ears; however, the number of CD4⁺ T cells was much higher than that of CD8⁺ T cells, indicating that CD4⁺ T cells are the main T cells that are increased in number during the development of Ni allergy (please refer to Figure 4 and the Source Data file).

14. (Figure 5) Can the authors provide the actual ear thickness data for the Sema3A cKO under the experimental conditions outlined in 5b like they do for 3e?

We have provided the actual ear thickness data of Sema3A fl/fl and Sema3A cKO mice in Figure 5. The related description in the figure legend has been revised.

15. (Figure 6) The authors again argue that numbers of cells are changing when comparing experimental groups but do not provide any absolute count data. Please provide absolute numbers.

We have redesigned the experiment and provided absolute numbers in revised Figure 6.

16. (Figure 6) Again, the F4/80/CD206 data seem to be driven by one animal per group (allergy 4 in Sema3A fl/fl and Allergy 3 in Sema3A cKO). Can the authors revisit the analysis of these data?

We have redesigned the flow cytometry experiment, and the new data are shown in Figure 6. We have also accordingly revised the manuscript text.

17. (Figure 6) The authors suggest that there is an increase in the M2 macrophage population compared to the M1 population and that this provides an anti-inflammatory environment. Did the authors measure anti-inflammatory cytokine levels?

We have measured the relative expression levels of anti-inflammatory cytokines by quantitative RT-PCR and the result has been added to Figure 5e. The corresponding manuscript text, figure legend, and Source Data file have been revised.

18. (Figure 6) Additionally, what happens to TSLP in the knockout model?

After careful consideration, we believe that TSLP is not directly related to the aim of this research, and hence, we decided to remove the contents about TSLP.

19. (Figure 6) Additionally, what changes take place with respect to infiltrating immune cell populations? Can the authors provide data similar to what is provided in Figure 4 for the WT animals? In the discussion the authors claim that *Sema3A* leads to *CXCL1* expression and that this is responsible for immune cell trafficking to the ear. In the cKO experiments, the authors do not show how the immune cell composition in the ears changes. They explore macrophage populations but do not expand on other cell types. These data would strengthen their argument.

We did perform a similar FACS analysis as in Figure 4, and the results are shown below. When comparing the ratio of immune cell number in the allergy group to that in the control group, there was no difference between the *Sema3A* cKO mice and control mice.

According to your suggestions, we have redesigned the experiments and revised the manuscript based on the new findings. Please refer to the revised manuscript.

20. (Discussion) The sentence at lines 204-207 is unclear. Please revise.

Since we have revised the discussion, the sentence at lines 204-207 has been deleted.

21. (Minor comments) Figure 1 legend has SMA3A instead of SEMA3A. Please correct the typo.

We have corrected the word to *Sema3A*.

22. (Minor comments) Also, I suggest changing the figure titles to statements to reflect the results described.

We have revised the figure titles to statements that describe the results.

23. (Minor comments) The statistical comparison scheme in Figure 1c and elsewhere (a, b, c, etc) is not immediately intuitive and should be described in the figure legends and not be only available in the source data files.

We have changed the statistical comparison scheme in Figure 1 to the same as that in other figures.

24. (Minor comments) Control GAPDH bands for some of the blots described in Figure 1c are suboptimal and likely difficult to interpret. Suggest choosing another protein to normalize against.

We have performed the western blotting assays again using β -actin as the internal control. Accordingly, the figures and legends have been revised.

25. (Minor comments) Also, why is Sema3A in some blots represented by two bands?

The antibody we used for Sema3A western blotting is a polyclonal antibody. We performed the western blotting assays again using a Sema3A monoclonal antibody and β -actin as the internal control. Accordingly, the results section and figure legends have been revised.

Reviewer #2:

Before responding to the comments and requests, we would like to first thank you for all your helpful suggestions, which have greatly benefitted the revised manuscript. Our responses to each of your concerns are listed below; we hope that you find them satisfactory. All changes made to the manuscript are indicated using red font in the manuscript file.

Major points:

1) In figure 2 the authors use siRNA to knock down the nickel-induced upregulation of Sema3A on PAM2.12. Using this setup, the author finds reduced TNF α production and p38 phosphorylation induced by nickel stimulation. However, they do not show the mechanism behind this – does Sema3A work in an autocrine manner on the KC? Please show the receptor expression on the KC (in general, it would be nice if the authors included the receptor expression on different cellular subsets found in the skin during the response to nickel). To further confirm that Sema3A can stimulate the KC, please stimulate the KC with either soluble Sema3A or do blocking experiments.

As shown in the figure below, we performed quantitative RT-PCR on soluble Sema3A (100 ng/ml)-stimulated keratinocytes. The result showed a decreased mRNA expression of Sema3A.

Meanwhile, we confirmed the expression of Sema3A receptors, Plexin A1 and Neuropilin 1, on both keratinocytes and bone marrow-derived DCs by PCR.

In addition, we have found several studies that support the idea that Sema3A functions in an autocrine manner (please see the references below).

- Bagci, T., Wu, J. K., Pfannl, R., Ilag, L. L., Jay, D. G. Autocrine semaphorin 3A signaling promotes glioblastoma dispersal. *Oncogene*. **28**, 3537-3550 (2009).
- Pan, H., Wanami, L. S., Dissanayake, T. R., Bachelder, R. E. Autocrine semaphorin3A stimulates alpha2 beta1 integrin expression/function in breast tumor cells. *Breast Cancer Res. Treat.* **118**, 197-205 (2009).
- Yamada, D., Takahashi, K., Kawahara, K., Maeda, T. Autocrine Semaphorin3A signaling is essential for the maintenance of stem-like cells in lung cancer. *Biochem. Biophys. Res. Commun.* **480**, 375-379 (2016).

2)In figure 3 the author shows that IL-17, IL-23, CCL20 and CXCL1 are all up-regulated in their nickel-model. In figure 5 the author shows that IL-1beta, TNFa, IL-23 and CXCL1 are all reduced in Sema3A cKO mice compared to WT mice during the response to nickel. Please show all cytokine/chemokines in both figure 3 and 5.

We have performed the quantitative RT-PCR again and attached the data in Figures 3 and 5. The IL-17 amplification showed a Ct value of approximately 35 in the quantitative RT-PCR, which was considered inaccurate, and hence, we have not included it in the revised manuscript.

3)Nickel allergy is a T cell mediated diseases and Sema3A is known to play a role in T cell activation/function. Still, the authors focus on the effect of Sema3A on the macrophage phenotype. Please include data showing the effect of Sema3A on the T cell response induced by nickel.

We have checked the T cells in this model and the result is as shown below. The figure shows the flow cytometry gating strategy for CD4⁺ T cells and CD8⁺ T cells.

As shown in the figure below, both CD4⁺ T cell and CD8⁺ T cell numbers were found to be increased in the Ni allergy-induced mouse ears; CD4⁺ T cells were the main T cells in the allergic tissue. In the cKO mice, there was a significant increase in the T cell number than that in the control mice. However, the CD4⁺/CD8⁺ ratio change in the allergy group between the two types of mice did not show any significant difference. The manuscript has been revised according to the new results.

Minor point:

The manuscript needs to be carefully checked for spelling mistakes and the

authors need to ensure that it is written in a correct English.

Thank you for your kind reminder. We have availed the services of Editage for professional English editing and proofreading.

Reviewer #3:

Before responding to the comments and requests, we would like to first thank you for all your helpful suggestions, which have greatly benefitted the revised manuscript. Our responses to each of your concerns are listed below; we hope that you find them satisfactory. All changes made to the manuscript are indicated using red font in the manuscript file.

Major comments:

1. They compared the between PBS and Ni challenge in sensitized mice. These data do not exclude the possibility of toxicity from Nickel. If you clarify the involvement of Sema3A in Ni allergy, not toxicity, you should compare the between sensitized and unsensitized mice.

We have compared the outcome of PBS and Ni challenge in both sensitized and unsensitized mice for three different genotypes. The ear thickness was not increased in the unsensitized mice. The data have been attached in Supplementary Figure 4.

2. They should clarify the Sema3A effect at the sensitization phase or effector phase.

In our study, we used the knockout mouse in which Sema3A was specifically knocked out from keratinocytes. When injecting Ni into the ear tissue as a re-challenge, the Ni allergy reaction was alleviated compared to that in the control mice. Therefore, we speculate that Sema3A promotes the process of Ni allergy in the effector phase. However, regarding the role of Sema3A in the sensitization phase, we still have insufficient evidence to draw a conclusion.

3. They clarified that Ni-induced Sema3A mediates TNF production and MAPK activation in vitro. But, in vivo data are insufficient to prove the Sema3A activates TNF and MAPK in Ni allergy.

Unfortunately, we have not checked the relationship between Sema3A and TNF/ MAPK in vivo. However, there are previous studies (shown below) that have demonstrated that Sema3A activates TNF and MAPK in inflammation in vivo.

Serum levels of Sema3A and TNF- α were all higher in patients with ankylosing spondylitis than in normal controls. The serum level of Sema3A was demonstrated to be significantly positively correlated with that of TNF- α .

- Liao, H. T., Lin, Y. F., Chou, C. T., et al. Semaphorin 3A in ankylosing spondylitis. *J. Microbiol. Immunol. Infect.* **52**, 151-157 (2019).

Sema3A treatment suppressed TNF- α release in the conjunctiva of EAC mice.

- Tanaka, J., Tanaka, H., Mizuki, N., et al. Semaphorin 3A controls allergic and inflammatory responses in experimental allergic conjunctivitis. *Int. J. Ophthalmol.* **8**, 1 (2015).

Sema3A in growth cones elicited rapid and differential activation of p38.

- Campbell, D. S., Holt, C. E. Apoptotic pathway and MAPKs differentially regulate chemotropic responses of retinal growth cones. *Neuron* **37**, 939-952 (2003).

A selective recruitment of MAPK during Sema3A-dependent migration or apoptosis was confirmed in native cerebellar progenitors.

- Bagnard, D., Sainturet, N., Meyronet, D., et al. Differential MAP kinases activation during semaphorin3A-induced repulsion or apoptosis of neural progenitor cells. *Mol. Cell. Neurosci.* **25**, 722-731 (2004).

Intravitreally injected Sema3A significantly inhibited the phosphorylation of p38MAPK signaling pathways.

- Yu, W., Bai, Y., Han, N., et al. Inhibition of pathological retinal neovascularization by semaphorin 3A. *Mol. Vision* **19**, 1397 (2013).

Sema3A levels and the activated form of p38 were elevated 3 days post-axotomy in a rat model of complete transection of the optic nerve.

- Nitzan, A., Kermer, P., Shirvan, A., et al. Examination of cellular and molecular events associated with optic nerve axotomy. *Glia* **54**, 545-556 (2006).

4. In vitro data, they explain that the nickel concentration (250uM) was determined from reference 11. However, in reference 11, they used 10uM Ni because of over 100uM affect cell viability. They should explain why they use 250uM Ni.

In the study cited under reference 11, 500 μ M Ni was demonstrated to affect cell viability. In our study, we wanted to verify the effect of Ni on cells at a high but safe concentration, and 250 μ M seemed reasonable. We have tested the viability of Pam2.12 cells after stimulation with 250 μ M Ni for different durations; the results are shown in Figure 1. The results showed that a 250 μ M concentration of Ni does not affect the cell viability.

5. In the previous data, they clarified the involvement of TSLP in Ni allergy (reference 10). You should state the effect of Ni-induced Sema3A on TSLP.

After careful consideration, we believe that TSLP is not directly related to the aim of this research, and hence, we decided to remove the contents about TSLP.

6. In flow cytometry data, the facs plot is unreliability, they should be reconsidered the experimental methods or add the isotype control.

We have added the isotype control in the newly designed flow cytometry assay; the gating strategy is shown in Figure 4. The isotype control is shown below.

Material and Method:

1. They should add the gating strategy in flow cytometry.

The newly designed gating strategy for flow cytometry is shown in revised Figure 4.

2. They should add the Fc block and dead cell staining in flow cytometry.

We have added the Fc block and dead cell staining in the newly designed flow cytometry assay; the gating strategy is shown in revised Figure 4 and described in the Methods.

3. They should add the cell numbers of Pam212.

The cell number of Pam2.12 used was 1×10^7 cells/10 cm dish. This information has been added to the Methods section.

Result:

1. In Figure 1C, they should show adequate statistical analysis.

We have revised the statistical analysis in Figure 1c.

2. In Figure 1, They should add the data of cell viability and explain the Sema3A production is due to cell death or not.

We have attached the cell viability results in Figure 1; the data show that a 250 μM concentration of Ni does not affect the cell viability. This indicates that the Sema3A production was not due to cell death.

3. In Figure 2A, they should state the time in control and Mock.

The samples of the Control and Mock groups were collected at 48 h; we have described this in the figure legend.

4. In Figure 2A, why they compare between control-control to mock-nickel?

We performed multiple comparisons and showed all the differences between any groups in the earlier manuscript. However, in the revised manuscript, we have revised the results and only show the meaningful comparisons.

5. In Figure 3C, the histological image show the ear thickness of Ni is 1.2mm, but in Figure 3E, the ear thickness of Ni is only 0.45mm. Why does it make such a difference?

As shown in the picture below, we performed the injection and measurement in the white spot in the center of the ear, and the allergic inflammation reactions are expected to occur inside the blue circle. For preparing sections, we collected the whole ear tissue and cut horizontally from the bottom of the ear to the top (direction indicated by the black arrow in the picture). The lower the section position is, the greater the measured ear thickness would be.

6. In Figure 3D, Isotype control are too dyed and should take adequate control.

As shown in the image below, taken at a lower magnification (200×), the cartilage(Δ), muscle, and fatty tissue (\bigcirc) would stain green as the background.

The isotype control images from both the control ear tissue and Ni-challenged ear tissue are shown below.

(isotype control image from control ear)

(isotype control image from Ni-challenged ear)

7. In Figure 3, they determined the involvement of Ni-induced Sema3A to TNF and MAPK activation. Why they do not check the these?

Unfortunately, we have not checked the relationship between Sema3A and TNF/ MAPK in vivo. However, there are previous studies (indicated below) that have proved that Sema3A activates TNF and MAPK in inflammation in vivo.

Serum levels of Sema3A and TNF- α were all higher in patients with ankylosing spondylitis than in normal controls. The serum level of Sema3A was demonstrated to be significantly positively correlated with that of TNF- α .

- Liao, H. T., Lin, Y. F., Chou, C. T., et al. Semaphorin 3A in ankylosing spondylitis. *J. Microbiol. Immunol. Infect.* **52**, 151-157 (2019).

Sema3A treatment suppressed TNF- α release in the conjunctiva of EAC mice.

- Tanaka, J., Tanaka, H., Mizuki, N., et al. Semaphorin 3A controls allergic and inflammatory responses in experimental allergic conjunctivitis. *Int. J. Ophthalmol.* **8**, 1 (2015).

Sema3A in growth cones elicited rapid and differential activation of p38.

- Campbell, D. S., Holt, C. E. Apoptotic pathway and MAPKs differentially regulate chemotropic responses of retinal growth cones. *Neuron* **37**, 939-952 (2003).

A selective recruitment of MAPK during Sema3A-dependent migration or apoptosis was confirmed in native cerebellar progenitors.

- Bagnard, D., Sainturet, N., Meyronet, D., et al. Differential MAP kinases activation during semaphorin3A-induced repulsion or apoptosis of neural progenitor cells. *Mol. Cell. Neurosci.* **25**, 722-731 (2004).

Intravitreally injected Sema3A significantly inhibited the phosphorylation of p38MAPK signaling pathways.

- Yu, W., Bai, Y., Han, N., et al. Inhibition of pathological retinal neovascularization by semaphorin 3A. *Mol. Vision* **19**, 1397 (2013).

Sema3A levels and the activated form of p38 were elevated 3 days post-axotomy in a rat model of complete transection of the optic nerve.

- Nitzan, A., Kermer, P., Shirvan, A., et al. Examination of cellular and molecular events associated with optic nerve axotomy. *Glia* **54**, 545-556 (2006).

8. In Figure 4 A,D,G, macrophages in the ear are a well-separated cell population, the staining and experimental methods should be reconsidered.

We only adapted two antibodies to identify macrophages; hence, we could not obtain a well-separated cell population. However, it was obvious that there was an increasing

trend in the number of macrophages in the Ni allergy group compared to that in the control group. We have re-designed the gating strategy and the new protocol is shown in Figure 4.

9. In Figure 4 B,E,H, they should show the cell numbers.

We have re-designed the experiment and the new gating strategy and data with cell numbers are shown in revised Figure 4.

10. In Figure 5B, they should show the data by ear thickness (mm).

We have added the data for ear thickness in Figure 5a (previously Figure 5b).

11. In Figure 6, they should state the time after Ni challenge.

All the samples *in vivo* were collected at 48 h after Ni challenge; we have clarified this in the figure legends.

12. In Figure 6A, they defined M1 and M2 macrophage by F4/80 and CD206.

This is not sufficient, and the cells should be firmly stained with the M1 marker.

We have re-designed this experiment and the new data are shown in Figure 6. We have also revised the manuscript text according to the new results.

13. In Figure 6A, M1/M2 population is 4.18%/1.94% in Sema3A fl/fl mice and 4.8% / 2.54% in Sema3A cKO mice. Is this the correct way to present the data in Figure 6C?

We have re-designed this experiment; the new data are shown in Figure 6. We have also revised the manuscript text according to the new results.

Figure legend:

1. They should add the sample size.

We have added the sample size in the figure legends.

2. In figure 1A, they need to modify Sma3A to Sema3A.

We have corrected the word to *Sema3A* in the Figure 1a legend.

Reviewers' comments:

Reviewer #1 (Remarks to the Author):

I would like to thank the authors for such a thorough revision of their work taking into account all comments and performing additional experiments to address the comments.

Please see below for minor comments:

Figure 3: The control I believe should be the Ni-challenged isotype treated ear for the Ni-challenged ear and the PBS-challenged for the control ear. So technically there should be two control images. It seems that the background staining is different in the two conditions.

Reviewer #2 (Remarks to the Author):

Thank you for a nice revision.

Reviewer #3 (Remarks to the Author):

No further comments to Authors.

Point-by-point responses to reviewer comments:

Reviewer #1:

Thank you so much for your helpful suggestion, we have added the two isotype control images for both Ni-challenged group and PBS-challenged control group in Figure 3e (as shown below), and the change in manuscript is highlighted, we hope that you find them satisfactory.

As for the different background staining in two controls, it is because that in the PBS control group, the ear tissue is not as swollen as in the Ni-challenged group.

Therefore, in the PBS control group, it showed non-specific green staining in the cartilage (Δ), muscle and fatty tissue (\bigcirc); however, in the Ni-challenged group, since the ear tissue is swollen, it would not include the cartilage, muscle and fatty tissue in the field of same magnification with control group, thus showing no non-specific green staining. Please check the examples below, the yellow square (\square) represents the magnified area.

(40 \times , PBS group; cartilage, muscle and fatty tissue included in the magnified zone)

(40 \times , Ni group; cartilage, muscle and fatty tissue NOT included in the magnified zone)